# Image Restoration via Diffusion Models with Dynamic Resolution

**Yang Zheng** [1]  **Wen Li** [1]  **Zhaoqiang Liu** [1]

## Abstract

Diffusion models (DMs) have exhibited remarkable efficacy in various image restoration tasks. However, existing approaches typically operate within the high-dimensional pixel space, resulting in high computational overhead. While methods based on latent DMs seek to alleviate this issue by utilizing the compressed latent space of a variational autoencoder, they require repeated encoder-decoder inference. This introduces significant additional computational burdens, often resulting in runtime performance that is even inferior to that of their pixel-space counterparts. To mitigate the computational inefficiency, this work proposes projecting data into lower-dimensional subspaces using dynamic resolution DMs to accelerate the inference process. We first fine-tune pretrained DMs for dynamic resolution priors and adapt DPS and DAPS, which are two widely used pixel-space methods for general image restoration tasks, into the proposed framework, yielding methods we refer to as SubDPS and SubDAPS, respectively. Given the favorable inference speed and reconstruction fidelity of SubDAPS, we introduce an enhanced variant termed SubDAPS++ to further boost both reconstruction efficiency and quality. Empirical evaluations across diverse image datasets and various restoration tasks demonstrate that the proposed methods outperform recent DM-based approaches in the majority of experimental scenarios. The code is available at https://github.com/StarNextDay/SubDAPS.git.

## 1. Introduction

Image restoration problems aim to recover the underlying image from degraded observations. Applications of image restoration span diverse fields, including medical imaging (Chung et al., 2022a), compressed sensing (Bora et al., 2017), and super-resolution (Saharia et al., 2023). The measurement model of image restoration is typically formulated as follows (Foucart & Rauhut, 2013; Saharia et al., 2022):

$$\boldsymbol{y} = \mathcal{A}(\boldsymbol{x}_0^*) + \boldsymbol{\nu}, \tag{1}$$

where $\boldsymbol{x}_0^* \in \mathbb{R}^d$ denotes the underlying image, $\boldsymbol{y} \in \mathbb{R}^m$ represents the degraded observation, $\mathcal{A} : \mathbb{R}^d \to \mathbb{R}^m$ is the forward operator, and $\boldsymbol{\nu} \sim \mathcal{N}(\boldsymbol{0}, \sigma^2 \boldsymbol{I}_m)$ denotes the Gaussian noise vector.

Traditional approaches for image restoration often rely on handcrafted priors derived from domain knowledge. Generative models introduce new paradigms for image restoration by leveraging priors learned from large-scale datasets (Bora et al., 2017; Liu & Scarlett, 2020; Jalal et al., 2020; Liu et al., 2021; Daras et al., 2021; 2022; Liu & Han, 2022; Liu et al., 2022; Chen et al., 2025b). Among generative approaches for image restoration, methods based on pretrained diffusion models (DMs) exhibit superior performance. However, existing DM-based methods (Kawar et al., 2022; Chung et al., 2022b; Wang et al., 2023; Chung et al., 2023; Song et al., 2023; Janati et al., 2024; Wang et al., 2024; Zheng et al., 2026c;b) typically operate within the high-dimensional pixel space, which contains certain computational redundancy in the early sampling stages (Jing et al., 2022). Although latent DM based approaches aim to mitigate this inefficiency by performing diffusion sampling within the compressed latent space of a variational autoencoder (VAE), they necessitate repeated encoder-decoder inference (Rout et al., 2023; Song et al., 2024; Chung et al., 2024; Rout et al., 2024; Zhang et al., 2025a; Kim et al., 2025b), resulting in a total inference time even exceeding that of the corresponding pixel-space methods when employing the same number of sampling steps.

Dynamic resolution DMs have recently attracted research interest (Jing et al., 2022; Zhang et al., 2023; Liu et al., 2024; Zheng et al., 2026a). While both dynamic resolution and latent DMs avoid consistently proceeding in full-dimensional pixel spaces, the former offers the advantage of operating without reliance on VAEs, which incur time-consuming encoder-decoder inference. Furthermore, the prior work (Oh et al., 2005) indicates that dynamic resolution strategies facilitate the recovery of global structures and

---

[1]School of Computer Science and Engineering, University of Electronic Science and Engineering of China. Correspondence to: Zhaoqiang Liu <zqliu12@gmail.com>.

*Proceedings of the 43rd International Conference on Machine Learning*, Seoul, South Korea. PMLR 306, 2026. Copyright 2026 by the author(s).

improve convergence efficiency. Motivated by the potential of dynamic resolution to enhance computational efficiency and reconstruction quality, this paper proposes methods leveraging dynamic resolution DMs to address general image restoration tasks.

## 1.1. Related Work

We discuss related works in three categories, namely image restoration with diffusion models in full-dimensional pixel space, image restoration with latent diffusion models, and dynamic resolution diffusion models.

**Image restoration with diffusion models in full-dimensional pixel space:** Due to their superior distribution modeling capabilities, pre-trained DMs serve as effective tools for image restoration (Chung et al., 2023; Li et al., 2024; Janati et al., 2024; Dou & Song, 2024; Peng et al., 2024; Zhang et al., 2024; Janati et al., 2025; Chang et al., 2025; Dou et al., 2025; Zheng et al., 2025; Gutha et al., 2025; Chen et al., 2025a; Li & Wang, 2025; Xue et al., 2025a;b; Zhang et al., 2025b; Zheng et al., 2025; Li & Wang, 2026; Zheng et al., 2026d). DM-based methods recover the underlying image through various frameworks. For instance, DPS (Chung et al., 2023) employs Tweedie's formula to approximate the underlying image and subsequently integrates this approximation with gradient-based posterior corrections. ΠGDM (Hen et al., 2025) incorporates pseudo-inverse guidance into the diffusion sampling process to improve the quality of reconstruction. DDRM (Kawar et al., 2022) employs singular value decomposition of the linear forward operator to approximate the measurement-matching term within the spectral domain. DDNM (Wang et al., 2023) decomposes measurements into null-space and range-space components based on the pseudo-inverse of the linear forward operator to facilitate reconstruction. Both DDRM and DDNM are tailored for linear image restoration tasks. DiffPIR (Zhu et al., 2023) adopts a proximal update scheme to approximate the conditional posterior mean. RED-diff (Mardani et al., 2023) leverages variational inference by introducing a tractable surrogate distribution to approximate the true posterior. MGPS (Moufad et al., 2025) introduces a midpoint decomposition of the backward diffusion process, utilizing a variational Gaussian approximation at intermediate steps to balance the dynamics of the prior with the accuracy of the likelihood guidance. DAPS (Zhang et al., 2025a) decouples the diffusion sampling trajectory to enable early-stage error correction. AdaPS (Hen et al., 2025) adaptively scales the likelihood guidance for robust DM-based restoration. Although pixel-space DM-based approaches demonstrate efficacy in image restoration, performing full-dimensional inference during the early reconstruction stage leads to computational redundancy (Oh et al., 2005; Jing et al., 2022) and increased computational overhead.

**Image restoration with latent diffusion models:** Methods based on latent DMs for image restoration also gain research attention. PSLD (Rout et al., 2023) combines the sampling process of DPS with an objective tailored for latent DMs to achieve satisfactory results on linear image generation tasks. ReSample (Song et al., 2024) designs a hard-constrained optimization problem and employs a resampling scheme to improve reconstruction fidelity. LatentDAPS (Zhang et al., 2025a) is the latent variant of DAPS, and it also decouples the diffusion sampling trajectory to enable early-stage error correction. SILO (Raphaeli et al., 2025) and InverseCrafter (Hong et al., 2025) share the same motivation to avoid repeated encoding and decoding by encoding the forward operator. In particular, SILO train a task-specific latent encoder. InverseCrafter extends the idea of encoding the measurement to the video domain and avoids task-specific training in SILO by combining the encoders of VAE with activation functions. While methods based on latent DMs benefit computationally from inference in a lower-dimensional latent space, these methods typically require repeated inference using the encoder and decoder, resulting in total reconstruction times that even exceed those of their corresponding full-dimensional pixel-space counterparts.

**Dynamic resolution diffusion models:** Since the advent of DMs (Sohl-Dickstein et al., 2015; Ho et al., 2020), extensive research focuses on mitigating the computational overhead during inference incurred by the iterative sampling process. One line of works concentrates on minimizing the number of sampling steps to accelerate inference (Song et al., 2021a;b; Lu et al., 2022; 2025; Karras et al., 2022; Liu et al., 2023; Zhao et al., 2023; Geng et al., 2025; Zheng et al., 2026a). Another line of works targets reducing per-step computational costs (Ho et al., 2022; Skorokhodov et al., 2024; Ma et al., 2024; Tian et al., 2025; Jeong et al., 2025).

Among methods aiming to reduce per-step computational costs, approaches leveraging dynamic resolution DMs have recently gained attention. In contrast to latent DMs, which utilize auxiliary generative models such as VAEs to reduce the dimensionality of latent features and rely on their generative capacity, dynamic resolution DMs operate without requiring additional neural networks for dimensionality reduction. Instead, dynamic resolution DMs accelerate the sampling process by reducing the spatial resolution of latent features, typically during early timesteps where fine details are less important (Sun et al., 2026; Zheng et al., 2026a). Diverse strategies facilitate this resolution reduction. For instance, Subspace DMs (Jing et al., 2022) restrict the initial sampling steps to a low-dimensional subspace, thereby accelerating inference. UDPM (Abu-Hussein & Giryes, 2024) has a similar motivation to Subspace DMs. Both UDPM and Subspace DMs accelerate generation by performing the early denoising process in lower-dimensional subspaces. They differ in that UDPM focuses on design-

ing upsampling and downsampling operations within the discrete-time framework, whereas Subspace DMs formulates the diffusion denoising process in continuous time. DVDP (Zhang et al., 2023) decomposes the image into multiple orthogonal components and employs an architecture that accommodates variable dimensions in both the training and inference phases. DiMR (Liu et al., 2024) proposes a multi-resolution network to refine features across different scales. Fresco (Zheng et al., 2026a) unifies re-noising and global structure across stages to preserve both efficiency and fidelity. Motivated by the potential of dynamic resolution DMs, this work leverages them to address image restoration problems, thereby avoiding unnecessary computational overhead during early timesteps and improving reconstruction efficiency (Oh et al., 2005).

## 1.2. Contributions

The main contributions of this work are summarized as follows:

- We propose methods that apply dynamic resolution DMs to general image restoration tasks. We first fine-tune existing pre-trained pixel-space DMs for dynamic resolution priors and subsequently adapt two pixel-space methods, namely DPS and DAPS, to the dynamic resolution framework, resulting in the so-called SubDPS and SubDAPS methods. To the best of our knowledge, this work constitutes the first application of dynamic resolution DMs to general image restoration tasks.

- Motivated by the favorable reconstruction efficiency and quality of SubDAPS, we introduce several strategies to further enhance it, leading to a method we refer to as SubDAPS++. Specifically, we modify the noise injection strategy during the later timesteps to mitigate the disruption of the diffusion prior caused by stochastic noise injection. Furthermore, we incorporate a corrector step that operates without additional model inference to enhance reconstruction fidelity. Moreover, we replace the Langevin dynamics with a conjugate gradient method applicable to both linear and nonlinear tasks to accelerate the reconstruction process.

- We conduct comprehensive empirical evaluations across diverse image datasets and restoration tasks. The experimental results demonstrate that the proposed methods outperform recent DM-based approaches in both reconstruction fidelity and sampling efficiency in the most scenarios, validating the effectiveness of our proposed methods.

## 2. Preliminaries

This section presents the preliminaries of full-dimensional pixel-space DMs, dynamic resolution DMs and two widely used DM-based methods for general image restoration tasks, namely DPS and DAPS.

### 2.1. Full-Dimensional Diffusion Models

DMs operate within a dual-process framework consisting of a forward stochastic process and a reverse process. The forward stochastic process progressively corrupts data into noise, whereas the corresponding reverse process reconstructs the original data distribution from the noise distribution. The dynamics of the forward process are governed by the following stochastic differential equation (SDE) (Jing et al., 2022):

$$\mathrm{d}\boldsymbol{x}_t = \boldsymbol{F}(\boldsymbol{x}_t, t)\mathrm{d}t + \boldsymbol{G}(\boldsymbol{x}_t, t)\mathrm{d}\boldsymbol{w}_t, \qquad (2)$$

where $\boldsymbol{F} : \mathbb{R}^d \times \mathbb{R}_+ \to \mathbb{R}^d$ and $\boldsymbol{G} : \mathbb{R}^d \times \mathbb{R}_+ \to \mathbb{R}^{d \times d}$ denote the drift and diffusion coefficients respectively, and $\boldsymbol{w}_t \in \mathbb{R}^d$ is a standard Wiener process. In full-dimensional DMs, $\boldsymbol{F}(\boldsymbol{x}_t, t)$ and $\boldsymbol{G}(\boldsymbol{x}_t, t)$ in Eq. (2) typically simplify to

$$\boldsymbol{F}(\boldsymbol{x}_t, t) = f(t)\boldsymbol{x}_t, \quad \boldsymbol{G}(\boldsymbol{x}_t, t) = g(t)\boldsymbol{I}_d, \qquad (3)$$

where $f(t)$ and $g(t)$ denote the time-dependent drift and diffusion coefficients, respectively. Substituting these terms into Eq. (2) yields

$$\mathrm{d}\boldsymbol{x}_t = f(t)\,\boldsymbol{x}_t\,\mathrm{d}t + g(t)\,\mathrm{d}\boldsymbol{w}_t. \qquad (4)$$

Let $p_t$ denote the marginal distribution of $\boldsymbol{x}_t$ with $p_0 = p_{\mathrm{data}}$ being the data distribution. For the time interval $t \in [0, T]$, the transition kernel $\boldsymbol{x}_t | \boldsymbol{x}_0$ follows a Gaussian distribution $\mathcal{N}(\alpha_t \boldsymbol{x}_0, \sigma_t^2 \boldsymbol{I}_d)$. Here, $\alpha_t$ and $\sigma_t$ are differentiable, non-negative, monotonic functions with bounded derivatives, chosen such that the signal-to-noise ratio (SNR) $\alpha_t^2 / \sigma_t^2$ decreases over time $t$. Furthermore, the boundary conditions for $\alpha_t$ and $\sigma_t$ ensure that the marginal distribution at the terminal time $p_T$ approximates $\mathcal{N}(\boldsymbol{0}, \tilde{\sigma}^2 \boldsymbol{I}_d)$ for a certain constant $\tilde{\sigma} > 0$. To ensure that the SDE in Eq. (4) has the same conditional distribution for $\boldsymbol{x}_t | \boldsymbol{x}_0$, the coefficients $f(t)$ and $g(t)$ satisfy $f(t) = \frac{\mathrm{d}\log\alpha_t}{\mathrm{d}t}$ and $g^2(t) = \frac{\mathrm{d}\sigma_t^2}{\mathrm{d}t} - 2\frac{\mathrm{d}\log\alpha_t}{\mathrm{d}t}\sigma_t^2$ (Lu et al., 2022). As derived in (Song et al., 2021b), the SDE in Eq. (4) admits a corresponding time-reversed diffusion process specified by

$$\mathrm{d}\boldsymbol{x}_t = \left(f(t)\,\boldsymbol{x}_t - g^2(t)\,\nabla_{\boldsymbol{x}_t}\log p_t(\boldsymbol{x}_t)\right)\mathrm{d}t + g(t)\,\mathrm{d}\bar{\boldsymbol{w}}_t, \qquad (5)$$

where $\bar{\boldsymbol{w}}_t$ denotes a standard Wiener process in reverse time, and $\nabla_{\boldsymbol{x}_t}\log p_t$ represents the score function.

Alternatively, Song et al. (Song et al., 2021b) show that a deterministic process sharing the same marginal distributions is governed by the following ordinary differential equation (ODE):

$$\mathrm{d}\boldsymbol{x}_t = \left(f(t)\,\boldsymbol{x}_t - \tfrac{1}{2}g^2(t)\,\nabla_{\boldsymbol{x}_t}\log p_t(\boldsymbol{x}_t)\right)\mathrm{d}t. \qquad (6)$$

Consequently, numerical integration of this ODE enables data generation, provided that the unknown score function

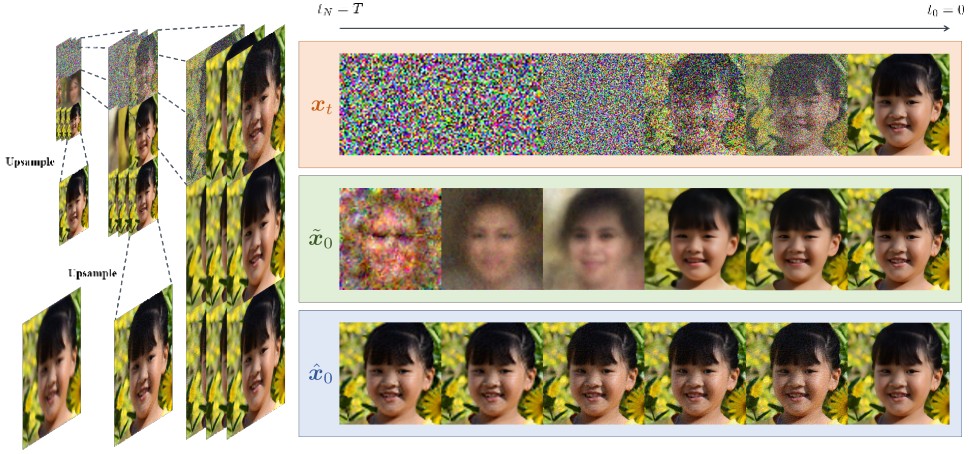

*Figure 1.* Overview of the proposed SubDAPS++ method. At each timestep $t_i$, the method utilizes the data prediction network $\boldsymbol{x_\theta}(\cdot, \cdot)$ to compute the unconditional estimate $\hat{\boldsymbol{x}}_0$. Subsequently, the conjugate gradient method aligns $\hat{\boldsymbol{x}}_0$ with the measurement $\boldsymbol{y}$, yielding the conditional estimate $\tilde{\boldsymbol{x}}_0$. If the dimensionality changes at timestep $t_i$, the algorithm upsamples $\tilde{\boldsymbol{x}}_0$ and injects random noise to match the diffusion prior. When $d_i = d$, the method either injects random noise or executes deterministic sampling, depending on the difference $\|\hat{\boldsymbol{x}}_0 - \tilde{\boldsymbol{x}}_0\|^2$. In our implementation, the resolution increases in two stages, from $64 \times 64 \times 3$ to $128 \times 128 \times 3$, and finally to $256 \times 256 \times 3$. It is observed that the model accurately recovers the global structure of the underlying image during early stages, while refining high-frequency details in later stages.

$\nabla_{\boldsymbol{x}_t} \log p_t(\boldsymbol{x}_t)$ is appropriately approximated. In practice, the score function $\nabla_{\boldsymbol{x}_t} \log p_t(\boldsymbol{x}_t)$ is typically parameterized as $(\alpha_t \boldsymbol{x_\theta}(\boldsymbol{x}_t, t) - \boldsymbol{x}_t)/\sigma_t^2$ (Kingma et al., 2021; Lu et al., 2025), where $\boldsymbol{x_\theta}(\boldsymbol{x}_t, t)$ represents a neural network trained to estimate the data.

### 2.2. Dynamic Resolution Diffusion Models

The forward process in Eq. (4) is specifically designed for full-dimensional DMs and is inapplicable to dynamic resolution DMs. To address this, the prior work (Jing et al., 2022) formulates a forward process for dynamic resolution DMs utilizing time intervals $0 = t_0 < t_1 < \cdots < t_N = T$. Specifically, within the interval $[t_{i-1}, t_i]$ for $1 \leq i \leq N$, the term $\boldsymbol{G}(\boldsymbol{x}_t, t)$ in Eq. (2) is set to:

$$\boldsymbol{G}(\boldsymbol{x}_t, t) = g(t) \boldsymbol{U}_i \boldsymbol{U}_i^{\mathrm{T}}, \qquad (7)$$

where $\boldsymbol{U}_i \in \mathbb{R}^{d \times d_i}$ consists of $d_i \leq d$ orthonormal columns that span a subspace of $\mathbb{R}^d$ and the dimensions associated with each interval satisfy the condition $d = d_0 \geq d_1 \geq \cdots \geq d_N$. Specifically, following (Jing et al., 2022), $\boldsymbol{U}_i$ is set as the upsampling matrix and thus $\boldsymbol{U}_i^T$ is the corresponding downsampling matrix. Over the entire time horizon, $\boldsymbol{F}(\boldsymbol{x}_t, t)$ in Eq. (2) is set to

$$\boldsymbol{F}(\boldsymbol{x}_t, t) = f(t)\boldsymbol{x}_t + \sum_{i=1}^{N} \delta(t - t_i)(\boldsymbol{U}_i \boldsymbol{U}_i^{\mathrm{T}} - \boldsymbol{I}_d)\boldsymbol{x}_t, \quad (8)$$

where $\delta(\cdot)$ denotes the Dirac delta function. We use $\dot{\boldsymbol{U}}_i := \boldsymbol{U}_{i-1}^{\mathrm{T}} \boldsymbol{U}_i \in \mathbb{R}^{d_{i-1} \times d_i}$ to denote the upsampling matrix between timesteps $t_i$ and $t_{i-1}$.

In contrast to full-dimensional DMs, which characterize the

reverse process as SDEs or ODEs and generate data by numerically solving these equations, dynamic resolution DMs typically generate data by iteratively executing the following three steps: 1) At timestep $t_i$, given $\boldsymbol{x}_{t_i} \in \mathbb{R}^{d_i}$, the sampling method first numerically solves the SDE or ODE from $t_i$ to $t_{i-1}$, analogous to the procedure in full-dimensional DMs, yielding an intermediate state. 2) Subsequently, the resolution of the intermediate state is increased from $d_i$ to $d_{i-1}$ via using the upsampling matrix $\dot{\boldsymbol{U}}_i$. As the dimensionality changes, this upsampled state deviates from the expected variance of the diffusion trajectory. 3) Consequently, an additional noise term is injected to ensure that the upsampled state aligns with the diffusion prior. Upon completion of this noise injection, the state $\boldsymbol{x}_{t_{i-1}} \in \mathbb{R}^{d_{i-1}}$ is obtained.

### 2.3. DPS and DAPS Methods

We introduce two widely used DM-based methods for general image restoration tasks, namely DPS (Chung et al., 2023) and DAPS (Zhang et al., 2025a), prior to detailing their adaptation from full-dimensional DMs to dynamic resolution DMs. DM-based methods for image restoration typically consider sampling from the following conditional SDE (Zhang et al., 2025a):

$$\mathrm{d}\boldsymbol{x}_t = \left( f(t)\,\boldsymbol{x}_t - g^2(t)\,\nabla_{\boldsymbol{x}_t} \log p_t(\boldsymbol{x}_t | \boldsymbol{y}) \right) \mathrm{d}t + g(t)\,\mathrm{d}\bar{\boldsymbol{w}}_t. \tag{9}$$

However, the conditional score function $\nabla_{\boldsymbol{x}_t} \log p_t(\boldsymbol{x}_t | \boldsymbol{y})$ is intractable. To address this, existing DM-based methods typically use Bayes' rule to decompose the conditional score into an unconditional score term and a measurement

consistency term (Daras et al., 2024):

$$\nabla_{\boldsymbol{x}_t} \log p_t(\boldsymbol{x}_t|\boldsymbol{y}) = \nabla_{\boldsymbol{x}_t} \log p_t(\boldsymbol{x}_t) + \nabla_{\boldsymbol{x}_t} \log p_t(\boldsymbol{y}|\boldsymbol{x}_t). \tag{10}$$

While the unconditional score $\nabla_{\boldsymbol{x}_t} \log p_t(\boldsymbol{x}_t)$ can be appropriately approximated by pre-trained DMs, the measurement consistency term $\nabla_{\boldsymbol{x}_t} \log p_t(\boldsymbol{y}|\boldsymbol{x}_t)$ remains computationally infeasible. To approximate this term, with the partition of timesteps $0 = t_0 < t_1 < \cdots < t_N = T$, methods generally first obtain a vector $\tilde{\boldsymbol{x}}_0^{t_i}$ that estimates the underlying data at timestep $t_i$ given current noisy state $\boldsymbol{x}_{t_i}$. Specifically, DPS employs the Tweedie's formula to obtain $\tilde{\boldsymbol{x}}_0^{t_i} = \boldsymbol{x}_{\boldsymbol{\theta}}(\boldsymbol{x}_{t_i}, t_i)$, whereas DAPS estimates $\tilde{\boldsymbol{x}}_0^{t_i}$ by numerically solving the unconditional ODE in Eq. (6) starting from $\boldsymbol{x}_{t_i}$ with a few steps. Subsequently, DPS approximates $\nabla_{\boldsymbol{x}_t} \log p_{t_i}(\boldsymbol{y}|\boldsymbol{x}_{t_i})$ as:

$$\nabla_{\boldsymbol{x}_t} \log p_{t_i}(\boldsymbol{y}|\boldsymbol{x}_{t_i}) \approx -\zeta_{t_i} \nabla_{\boldsymbol{x}_t} \left\| \boldsymbol{y} - \mathcal{A}(\tilde{\boldsymbol{x}}_0^{t_i}) \right\|^2, \tag{11}$$

where $\zeta_{t_i}$ denotes a time-dependent guidance strength parameter. With this approximation for the conditional score function, DPS adapts the DDPM sampling algorithm (Ho et al., 2020) to approximately solve the SDE in Eq. (9). On the other hand, DAPS first formulates an optimization problem as follows:

$$\hat{\boldsymbol{x}}_0^{t_i} = \arg\min_{\bar{\boldsymbol{x}}_0} \left( r_{t_i} \|\bar{\boldsymbol{x}}_0 - \tilde{\boldsymbol{x}}_0^{t_i}\|^2 + \|\boldsymbol{y} - \mathcal{A}(\bar{\boldsymbol{x}}_0)\|^2 \right), \tag{12}$$

where $r_{t_i}$ is a time-dependent regularization hyperparameter, and then approximately solves the optimization problem in Eq. (12) via Langevin dynamics. Subsequently, $\boldsymbol{x}_{t_{i-1}}$ is sampled from $\mathcal{N}(\alpha_{t_{i-1}} \hat{\boldsymbol{x}}_0^{t_i}, \sigma_{t_{i-1}}^2 \boldsymbol{I}_d)$. Starting from time $t_N = T$, DAPS approximately solves the SDE in Eq. (9) by repeating this process.

## 3. Methods

In this section, we first describe the training of dynamic resolution DMs via the fine-tuning of pre-trained DMs. We then detail the adaptation of two widely used DM-based methods, namely DPS and DAPS, from full-dimensional DMs to the dynamic resolution framework, resulting in our SubDPS and SubDAPS methods. Motivated by the inference speed and reconstruction fidelity of SubDAPS, we introduce an enhanced variant, denoted as SubDAPS++, to further improve reconstruction performance. Specifically, we modify the noise injection strategy during the final timesteps to mitigate the disruption of the diffusion prior caused by stochastic noise. Furthermore, we incorporate a corrector step that operates without additional model evaluations to enhance reconstruction fidelity. Finally, to accelerate the measurement consistency process, we replace the Langevin dynamics used in SubDAPS with a conjugate gradient method. Notably, we employ a first-order Taylor expansion to derive a closed-form line search solution applicable to general tasks, thereby enhancing computational efficiency.

### 3.1. Training of Dynamic Resolution Diffusion Models

Rather than training dynamic resolution DMs from randomly initialized parameters, we fine-tune pre-trained full-dimensional DMs (Dhariwal & Nichol, 2021; Chung et al., 2023), which inherently possess learned structural priors. We implement dynamic resolution across three distinct dimensions, denoted as $d > \tilde{d} > \hat{d}$, utilizing corresponding downsampling matrices $\tilde{U}^{\text{T}}$ and $\hat{U}^{\text{T}}$.[1] Adhering to the training strategy described in (Dhariwal & Nichol, 2021), we formulate the objective for dynamic resolution DMs as follows:

$$\arg\min_{\boldsymbol{\theta}} \mathbb{E}_{t \sim [0,T], \boldsymbol{x}_0 \sim p_0} \big[ \mathbb{E}_{\boldsymbol{\epsilon} \sim \mathcal{N}(\boldsymbol{0}, \boldsymbol{I}_d)} \| \boldsymbol{x}_0 - \boldsymbol{x}_{\boldsymbol{\theta}}(\alpha_t \boldsymbol{x}_0 + \sigma_t \boldsymbol{\epsilon}, t) \|^2$$
$$+ \mathbb{E}_{\boldsymbol{\epsilon} \sim \mathcal{N}(\boldsymbol{0}, \boldsymbol{I}_{\tilde{d}})} \| \tilde{U}^{\text{T}} \boldsymbol{x}_0 - \boldsymbol{x}_{\boldsymbol{\theta}}(\alpha_t \tilde{U}^{\text{T}} \boldsymbol{x}_0 + \sigma_t \boldsymbol{\epsilon}, t) \|^2$$
$$+ \mathbb{E}_{\boldsymbol{\epsilon} \sim \mathcal{N}(\boldsymbol{0}, \boldsymbol{I}_{\hat{d}})} \| \hat{U}^{\text{T}} \boldsymbol{x}_0 - \boldsymbol{x}_{\boldsymbol{\theta}}(\alpha_t \hat{U}^{\text{T}} \boldsymbol{x}_0 + \sigma_t \boldsymbol{\epsilon}, t) \|^2 \big]. \tag{13}$$

Training details are provided in Appendix C. Similar to the pre-training of pixel-space and latent-space DMs, the fine-tuning of the proposed method is performed only once. No further training is required for subsequent inference tasks.[2]

### 3.2. SubDPS and SubDAPS

We first illustrate the adaptation of DPS. Consider the partition of timesteps $0 = t_0 < \cdots < t_N = T$ and the corresponding upsampling matrices $\{\boldsymbol{U}_i\}_{i=1}^N$. At time $t_i$, if the dimension $d_{i-1} = d_i$, we can approximate the measurement consistency term for the dynamic resolution sampling process analogous to Eq. (11) as follows:

$$\nabla_{\boldsymbol{x}_t} \log p_{t_i}(\boldsymbol{y}|\boldsymbol{x}_{t_i}) \approx -\zeta_{t_i} \nabla_{\boldsymbol{x}_{t_i}} \|\boldsymbol{y} - \mathcal{A}(\boldsymbol{U}_i \boldsymbol{x}_{\boldsymbol{\theta}}(\boldsymbol{x}_{t_i}, t_i))\|^2. \tag{14}$$

Subsequently, the standard DDPM sampling algorithm can be employed to sample $\boldsymbol{x}_{t_{i-1}}$ following the original DPS scheme. However, when $d_{i-1} \neq d_i$, upsampling the state $\boldsymbol{x}_{t_{i-1}}$ requires modifications to align it with the diffusion prior. Motivated by DAPS, which suggests that random noise injection helps correct error accumulation at early timesteps, we forego explicit modification on $\boldsymbol{x}_{t_{i-1}}$ to match the diffusion prior. Instead, we generate $\boldsymbol{x}_{t_{i-1}}$ by directly injecting random noise into the upsampled data estimate $\dot{\boldsymbol{U}}_i \boldsymbol{x}_{\boldsymbol{\theta}}(\boldsymbol{x}_{t_i}, t_i)$ as follows:

$$\boldsymbol{x}_{t_{i-1}} = \alpha_{t_{i-1}} \dot{\boldsymbol{U}}_i \boldsymbol{x}_{\boldsymbol{\theta}}(\boldsymbol{x}_{t_i}, t_i) + \sigma_{t_{i-1}} \boldsymbol{\epsilon}_i, \ \boldsymbol{\epsilon}_i \sim \mathcal{N}(\boldsymbol{0}, \boldsymbol{I}_{d_{i-1}}). \tag{15}$$

This strategy facilitates the adaptation of general full-dimensional DM-based methods to the dynamic resolution

---

[1]In our experiments, we adopt the configuration suggested in (Jing et al., 2022), setting $d = 256 \times 256 \times 3$, $\tilde{d} = 128 \times 128 \times 3$, and $\hat{d} = 64 \times 64 \times 3$.

[2]The fine-tuned models are available at https://github.com/StarNextDay/SubDAPS.git. Additionally, our framework is compatible with off-the-shelf dynamic-resolution DMs or ensembles of resolution-specific pretrained DMs. We show corresponding results in Appendix B.

framework. The corresponding algorithm for the adaptation of DPS is referred to as SubDPS.

Next, we detail the adaptation of DAPS. According to Eq. (12), we formulate the loss function for the dynamic resolution process at $t_i$ as

$$\mathcal{L}(\boldsymbol{x}; \tilde{\boldsymbol{x}}_0^{t_i}, t_i) = r_{t_i} \|\boldsymbol{x} - \tilde{\boldsymbol{x}}_0^{t_i}\|^2 + \|\boldsymbol{y} - \mathcal{A}(\boldsymbol{U}_i\boldsymbol{x})\|^2, \quad (16)$$

where recall that $\tilde{\boldsymbol{x}}_0^{t_i}$ represents the estimate of the underlying data, obtained by numerically solving the ODE in Eq. (6) starting from $\boldsymbol{x}_{t_i}$ over multiple steps. After obtaining the estimate $\hat{\boldsymbol{x}}_0^{t_i}$ by approximately solving $\hat{\boldsymbol{x}}_0^{t_i} = \arg\min_{\bar{\boldsymbol{x}}_0} \mathcal{L}(\bar{\boldsymbol{x}}_0; \tilde{\boldsymbol{x}}_0^{t_i}, t_i)$ via Langevin dynamics, the state $\boldsymbol{x}_{t_{i-1}}$ at the subsequent timestep $t_{i-1}$ is derived as

$$\boldsymbol{x}_{t_{i-1}} = \alpha_{t_{i-1}} \dot{\boldsymbol{U}}_i \hat{\boldsymbol{x}}_0^{t_i} + \sigma_{t_{i-1}} \boldsymbol{\epsilon}_i, \ \boldsymbol{\epsilon}_i \sim \mathcal{N}(\boldsymbol{0}, \boldsymbol{I}_{d_{i-1}}). \quad (17)$$

We denote the adaptation of DAPS as SubDAPS. The complete procedures of SubDPS and SubDAPS are presented in Appendix D for clarity.

## 3.3. SubDAPS++

As our empirical findings suggest, SubDAPS leads to favorable inference speed and reconstruction fidelity for various image restoration tasks. Built upon SubDAPS, we propose the SubDAPS++ method, which uses several strategies to further enhance the reconstruction efficiency and quality. We first address the noise injection mechanism. SubDAPS injects random noise across all timesteps, as defined in Eq. (17). However, this continuous injection may disrupt the diffusion prior, potentially yielding unrealistic artifacts, particularly as the timestep approaches zero. To mitigate this, we propose identifying a cutoff timestep, after which random noise injection is ceased in favor of deterministic state calculation. We establish two criteria for this transition. First, we require the stabilization of dimensionality. Let $h = \min\{i \in \{1, \ldots, N\} \mid d_{i-1} \neq d_i\}$ be the index corresponding to the smallest timestep of a change in dimension. For all $i < h$, the dimensionality $d_i$ equals the full data dimensionality $d$. We then introduce a convergence condition to regulate the transition between stochastic and deterministic sampling. Specifically, for a given timestep $t_i$, if the squared difference between the unconditional prediction $\boldsymbol{x}_{\boldsymbol{\theta}}(\boldsymbol{x}_{t_i}, t_i)$ and the approximate solution $\hat{\boldsymbol{x}}_0^{t_i}$ that minimizes the loss in Eq. (16) is sufficiently small, we suppress noise injection. We quantify this proximity using a threshold parameter $\tau$. When $i < h$ and $\|\boldsymbol{x}_{\boldsymbol{\theta}}(\boldsymbol{x}_{t_i}, t_i) - \hat{\boldsymbol{x}}_0^{t_i}\|^2 \leq \tau$, we perform the update deterministically as follows:

$$\boldsymbol{x}_{t_{i-1}} = \alpha_{t_{i-1}} \hat{\boldsymbol{x}}_0^{t_i} + \frac{\sigma_{t_{i-1}}}{\sigma_{t_i}}(\boldsymbol{x}_{t_i} - \alpha_{t_i}\hat{\boldsymbol{x}}_0^{t_i}). \quad (18)$$

Otherwise, we maintain the noise injection strategy described in Eq. (17).

We also introduce a corrector step applied subsequent to the dynamic resolution DM sampling process to further improve reconstruction fidelity. Similar to that in UniPC (Zhao et al., 2023), this corrector step enhances the numerical accuracy without requiring additional neural network evaluations. Upon completing the sampling of $\boldsymbol{x}_{t_0}$, we obtain a denoising trajectory $\{\boldsymbol{x}_{t_i}\}_{i=0}^N$ comprising states at all timesteps. Adopting the predictor-corrector framework proposed in UniPC (Zhao et al., 2023), we refine this trajectory by correcting the state $\boldsymbol{x}_{t_{i-1}}$ to yield $\boldsymbol{x}_{t_{i-1}}^c$:

$$\boldsymbol{x}_{t_{i-1}}^c = \frac{\sigma_{t_{i-1}}}{\sigma_{t_i}} \dot{\boldsymbol{U}}_i \boldsymbol{x}_{t_i}^c - \left(\sigma_{t_{i-1}} \frac{\alpha_{t_i}}{\sigma_{t_i}} - \alpha_{t_{i-1}}\right) \hat{\boldsymbol{x}}_0^{t_{i-1}}$$
$$- \sigma_{t_{i-1}} \mathcal{I}_i \frac{\hat{\boldsymbol{x}}_0^{t_{i-1}} - \dot{\boldsymbol{U}}_i \hat{\boldsymbol{x}}_0^{t_i}}{\lambda_{t_{i-1}} - \lambda_{t_i}}, \quad (19)$$

where $\lambda_t := \log \frac{\alpha_t}{\sigma_t}$ denotes the half log-SNR and $\mathcal{I}_i := \int_{\lambda_{t_{i-1}}}^{\lambda_{t_i}} e^\lambda (\lambda - \lambda_{t_i}) \mathrm{d}\lambda$.

Additionally, compared with SubDAPS, SubDAPS++ utilizes $\boldsymbol{x}_{\boldsymbol{\theta}}(\boldsymbol{x}_{t_i}, t_i)$ instead of multi-step sampling to estimate the data prediction $\tilde{\boldsymbol{x}}_0^{t_i}$ at timestep $t_i$ for faster inference. To accelerate convergence, we also use the conjugate gradient method to minimize the loss in Eq. (16), as detailed in Section 3.4, and Algorithm 1 presents the complete procedure for the SubDAPS++ method.

## 3.4. Conjugate Gradient Method for SubDAPS++

We utilize conjugate gradient descent to minimize the loss in Eq. (16). For the $j$-th update of the conjugate gradient method at timestep $t_i$, the step size $\alpha_j$ is determined via a line search, formulated as

$$\alpha_j = \arg\min_\alpha \mathcal{L}(\bar{\boldsymbol{x}}_0^{(j)} + \alpha \boldsymbol{d}_j; \tilde{\boldsymbol{x}}_0^{t_i}, t_i), \quad (20)$$

where $\bar{\boldsymbol{x}}_0^{(j)}$ denotes the estimated vector of the $j$-th iteration and $\boldsymbol{d}_j$ represents the search direction. To efficiently solve the line search problem in Eq. (20), we linearize the operator $\mathcal{A}(\boldsymbol{U}_i(\bar{\boldsymbol{x}}_0^{(j)} + \alpha \boldsymbol{d}_j))$ via a first-order Taylor expansion around the current estimate $\bar{\boldsymbol{x}}_0^{(j)}$ as follows:

$$\mathcal{A}(\boldsymbol{U}_i(\bar{\boldsymbol{x}}_0^{(j)} + \alpha \boldsymbol{d}_j)) \approx \mathcal{A}(\boldsymbol{U}_i\bar{\boldsymbol{x}}_0^{(j)}) + \alpha \boldsymbol{\omega}_j, \quad (21)$$

where $\boldsymbol{\omega}_j := \nabla_{\bar{\boldsymbol{x}}_0^{(j)}} \mathcal{A}(\boldsymbol{U}_i\bar{\boldsymbol{x}}_0^{(j)}) \cdot \boldsymbol{d}_j$. Consequently, the line search problem in Eq. (20) admits the following closed-form solution:

$$\alpha_j = (\boldsymbol{g}_j^\mathrm{T} \boldsymbol{d}_j) / (r_{t_i} \boldsymbol{d}_j^\mathrm{T} \boldsymbol{d}_j + \boldsymbol{\omega}_j^\mathrm{T} \boldsymbol{\omega}_j), \quad (22)$$

where $\boldsymbol{g}_j = -\nabla_{\bar{\boldsymbol{x}}_0^{(j)}} \mathcal{L}(\bar{\boldsymbol{x}}_0^{(j)}; \tilde{\boldsymbol{x}}_0^{t_i}, \boldsymbol{y})$. Subsequently, we update $\bar{\boldsymbol{x}}_0^{(j)}$ and calculate $\boldsymbol{g}_{j+1}$ as follows:

$$\bar{\boldsymbol{x}}_0^{(j+1)} = \bar{\boldsymbol{x}}_0^{(j)} + \alpha_j \boldsymbol{d}_j, \quad (23)$$

$$\boldsymbol{g}_{j+1} = \nabla_{\bar{\boldsymbol{x}}_0^{(j+1)}} \mathcal{L}(\bar{\boldsymbol{x}}_0^{(j+1)}; \tilde{\boldsymbol{x}}_0^{t_i}, t_i). \quad (24)$$

**Algorithm 1** SubDAPS++

**Require:** Data prediction model $\boldsymbol{x}_{\boldsymbol{\theta}}(\cdot, \cdot)$, observation $\boldsymbol{y}$, forward operator $\mathcal{A}(\cdot)$, total number of sampling steps $N$ and inner iterations $J$, noise schedule $\{\sigma_{t_i}\}_{i=0}^N, \{\alpha_{t_i}\}_{i=0}^N$, regularization parameter $\{r_{t_i}\}_{i=1}^N$, upsampling matrices $\{\boldsymbol{U}_i\}_{i=1}^N$, upsampling timesteps $t_h$, noise level $\sigma$

1: Sample $\boldsymbol{x}_{t_N} \sim \mathcal{N}(\boldsymbol{0}, \sigma_{t_N}^2 \boldsymbol{I}_{d_N})$
2: **for** $i = N, \ldots, 1$ **do**
3: $\quad \tilde{\boldsymbol{x}}_0^{t_i} = \boldsymbol{x}_{\boldsymbol{\theta}}(\boldsymbol{x}_{t_i}, t_i)$
4: $\quad$ Obtain $\hat{\boldsymbol{x}}_0^{t_i}$ using the conjugate gradient descent method (detailed in Algorithm 4 in Appendix D)
5: $\quad$ **if** $\|\hat{\boldsymbol{x}}_0^{t_i} - \tilde{\boldsymbol{x}}_0^{t_i}\|^2 \geq \tau$ or $t_i \geq t_h$ **then**
6: $\qquad$ Sample $\boldsymbol{\epsilon}_i \sim \mathcal{N}(\boldsymbol{0}, \boldsymbol{I}_{d_{i-1}})$
7: $\qquad \boldsymbol{x}_{t_{i-1}} = \alpha_{t_{i-1}} \dot{\boldsymbol{U}}_i \hat{\boldsymbol{x}}_0^{t_i} + \sigma_{t_{i-1}} \boldsymbol{\epsilon}_i$
8: $\quad$ **else**
9: $\qquad \boldsymbol{x}_{t_{i-1}} = \alpha_{t_{i-1}} \hat{\boldsymbol{x}}_0^{t_i} + \frac{\sigma_{t_{i-1}}}{\sigma_{t_i}} (\boldsymbol{x}_{t_i} - \alpha_{t_i} \hat{\boldsymbol{x}}_0^{t_i})$
10: $\quad$ **end if**
11: **end for**
12: Set $\hat{\boldsymbol{x}}_0^{t_0} = \boldsymbol{x}_{t_0}$ and $\boldsymbol{x}_{t_N}^c = \boldsymbol{x}_{t_N}$
13: **for** $i = N, \ldots, 1$ **do**
14: $\quad \mathcal{I}_i = \int_{\lambda_{t_{i-1}}}^{\lambda_{t_i}} e^\lambda (\lambda - \lambda_{t_i}) \mathrm{d}\lambda$
15: $\quad \boldsymbol{x}_{t_{i-1}}^c = \frac{\sigma_{t_{i-1}}}{\sigma_{t_i}} \dot{\boldsymbol{U}}_i \boldsymbol{x}_{t_i}^c - \left(\sigma_{t_{i-1}} \frac{\alpha_{t_i}}{\sigma_{t_i}} - \alpha_{t_{i-1}}\right) \hat{\boldsymbol{x}}_0^{t_{i-1}} - \sigma_{t_{i-1}} \mathcal{I}_i \frac{\hat{\boldsymbol{x}}_0^{t_{i-1}} - \dot{\boldsymbol{U}}_i \hat{\boldsymbol{x}}_0^{t_i}}{\lambda_{t_{i-1}} - \lambda_{t_i}}$
16: **end for**
17: **Return** $\boldsymbol{x}_{t_0}^c$

We then update the search direction using $\boldsymbol{g}_{j+1}$. As the Fletcher-Reeves method is effective when the line search is inexact (Rivaie et al., 2015; Jiang & Jian, 2019), we adopt it to update the search direction, leading to:

$$\boldsymbol{d}_{j+1} = \boldsymbol{g}_{j+1} + \frac{\boldsymbol{g}_{j+1}^{\mathrm{T}} \boldsymbol{g}_{j+1}}{\boldsymbol{g}_j^{\mathrm{T}} \boldsymbol{g}_j} \boldsymbol{d}_j. \tag{25}$$

The complete conjugate gradient descent procedure for the image restoration process is summarized in Algorithm 4 in Appendix D for clarity.

## 4. Experimental Results

We evaluate our proposed methods on two $256 \times 256$ validation datasets, namely FFHQ (Karras et al., 2019) and ImageNet (Deng et al., 2009). Following the experimental settings of recent DM-based methods for image restoration (Zhu et al., 2023; Zhang et al., 2025a), we sample 100 images from the validation sets for evaluation. To quantify reconstruction quality and assess the perception-distortion trade-off (Blau & Michaeli, 2018), we report both distortion and perceptual metrics. Specifically, we employ the peak signal-to-noise ratio (PSNR) and structural similarity index measure (SSIM) to measure distortion, alongside the

learned perceptual image patch similarity (LPIPS) (Zhang et al., 2018) and Fréchet inception distance (FID) (Heusel et al., 2017) for measuring perceptual quality. **Bold** and underlined values indicate the best and second-best performance, respectively.

In Section 4, we evaluate the proposed methods against recent DM-based approaches across four linear tasks and two nonlinear tasks. Additionally, we report computational costs with respect to runtime and memory usage in Section 4.2. We also investigate the impact of the threshold parameter $\tau$ and assess the efficacy of the corrector within SubDAPS++, presenting these results in Appendix H.

### 4.1. Main Experiments

We evaluate our proposed methods across image restoration tasks involving different operators. We conduct evaluations on four linear tasks, namely inpainting (random 70%), super-resolution, Gaussian deblurring and motion deblurring and two nonlinear tasks including nonlinear deblurring and high dynamic range recovery. For all tasks, we compare the proposed SubDPS, SubDAPS, SubDAPS++ methods with baseline DM-based approaches, including DPS (Chung et al., 2023), ΠGDM (Song et al., 2023)0, DiffPIR (Zhu et al., 2023), RED-diff (Mardani et al., 2023), MGPS (Moufad et al., 2025), AdaPS (Hen et al., 2025), and DAPS (Zhang et al., 2025a).[3] We also compare our methods with two methods based on latent DMs, namely ReSample (Song et al., 2024) and LatentDAPS (Zhang et al., 2025a).[4] Except for DPS and SubDPS, which require 1000 neural function evaluations (NFEs) to attain reasonably good reconstructions, all methods employ 100 NFEs. All tasks settings follow prior works (Wang et al., 2024; Zhang et al., 2025a) and are detailed in Appendix E. To ensure a fair comparison, all methods utilize the same fine-tuned DMs, with the exception of methods based on latent DMs.[5] In all experiments, the proposed SubDAPS++ method implements resolution transitions at two specific timestep indexes, $s = \lfloor 2N/3 \rfloor$ and $h = \lfloor N/3 \rfloor$. Specifically, the resolution increases from $64 \times 64 \times 3$ to $128 \times 128 \times 3$ at timestep $t_s$, and subsequently to $256 \times 256 \times 3$ at timestep $t_h$. For other timesteps, the dimensionality remains constant. We perform all experiments under Gaussian noise with a noise level of $\sigma = 0.05$ using a single NVIDIA GeForce RTX 4090 GPU. The settings of tasks and hyper-parameters on each task are detailed in the Appendix E.

---

[3]For AdaPS, we use the variant based on ΠGDM (Song et al., 2023).

[4]Additionally, we examine a variant of SubDAPS++ that operates at the full dimensionality, referred to as SubDAPS-F++. Corresponding experimental results are presented in Appendix I.

[5]With the exception of methods based on latent DMs, all methods utilize the same fine-tuned DMs, as the underlying U-Net architecture inherently accommodates dynamic resolution inputs.

*Table 1.* Quantitative evaluation of four linear and two nonlinear tasks under Gaussian noise ($\sigma = 0.05$), using 100 validation images from the FFHQ and ImageNet datasets.

| | | **FFHQ** (256 × 256) | | | | **ImageNet** (256 × 256) | | | | **FFHQ** (256 × 256) | | | | **ImageNet** (256 × 256) | | | |
|---|---|---|---|---|---|---|---|---|---|---|---|---|---|---|---|---|---|
| | | **Inpainting (Random 70%)** | | | | | | | | **Super-Resolution** (4×) | | | | | | | |
| Methods | Type | PSNR↑ | SSIM↑ | LPIPS↓ | FID↓ | PSNR↑ | SSIM↑ | LPIPS↓ | FID↓ | PSNR↑ | SSIM↑ | LPIPS↓ | FID↓ | PSNR↑ | SSIM↑ | LPIPS↓ | FID↓ |
| DPS | *Pixel* | 28.54 | 0.823 | 0.116 | 62.53 | 25.33 | 0.687 | 0.299 | 141.99 | 24.00 | 0.680 | 0.191 | 82.89 | 21.68 | 0.525 | 0.432 | 212.64 |
| ΠGDM | *Pixel* | 32.07 | 0.910 | 0.068 | 55.59 | 28.18 | 0.836 | 0.139 | 73.91 | 27.85 | 0.808 | 0.101 | 70.00 | 23.11 | 0.622 | 0.215 | 104.39 |
| DiffPIR | *Pixel* | 32.16 | 0.901 | 0.052 | **41.73** | **28.70** | 0.840 | **0.092** | **48.71** | 27.64 | 0.775 | 0.116 | 67.96 | 24.31 | 0.622 | 0.365 | 131.18 |
| RED-diff | *Pixel* | 29.54 | 0.828 | 0.110 | 77.00 | 26.67 | 0.766 | 0.173 | 93.24 | 27.71 | 0.717 | 0.333 | 122.28 | 24.63 | 0.612 | 0.556 | 171.58 |
| MGPS | *Pixel* | 31.41 | 0.894 | **0.050** | 42.15 | 28.25 | 0.828 | 0.105 | 51.14 | 27.58 | 0.792 | 0.110 | 62.00 | 25.00 | 0.680 | 0.304 | 122.34 |
| DAPS | *Pixel* | 30.68 | 0.835 | 0.073 | 56.73 | 27.63 | 0.780 | 0.115 | 56.73 | 28.88 | 0.806 | 0.162 | 74.16 | 25.54 | 0.691 | 0.354 | 122.32 |
| AdaPS | *Pixel* | **32.34** | **0.917** | 0.057 | 48.65 | 28.40 | **0.846** | 0.115 | 60.74 | 27.34 | 0.786 | **0.090** | 60.73 | 24.28 | 0.679 | **0.194** | **93.85** |
| Resample | *Latent* | 28.88 | 0.835 | 0.099 | 83.78 | 25.23 | 0.723 | 0.189 | 105.68 | 23.64 | 0.511 | 0.403 | 144.18 | 21.76 | 0.433 | 0.440 | 164.58 |
| LatentDAPS | *Latent* | 31.17 | 0.899 | 0.090 | 69.10 | 27.33 | 0.810 | 0.164 | 85.24 | 28.56 | 0.829 | 0.174 | 70.46 | 25.43 | 0.714 | 0.377 | 137.51 |
| **SubDPS** | *Dynamic* | 28.13 | 0.812 | 0.128 | 69.78 | 24.81 | 0.661 | 0.338 | 162.62 | 23.74 | 0.672 | 0.207 | 82.25 | 21.50 | 0.521 | 0.510 | 227.29 |
| **SubDAPS** | *Dynamic* | 30.71 | 0.835 | 0.073 | 57.31 | 27.62 | 0.780 | 0.114 | 56.20 | 28.89 | 0.806 | 0.161 | 74.85 | 25.55 | 0.691 | 0.354 | 119.93 |
| **SubDAPS++** | *Dynamic* | 32.21 | 0.907 | 0.056 | 43.15 | 28.61 | 0.843 | **0.092** | 49.15 | 29.34 | 0.838 | 0.157 | 64.61 | 25.79 | 0.722 | 0.358 | 113.91 |
| | | **Gaussian Deblurring** | | | | | | | | **Motion Deblurring** | | | | | | | |
| Methods | Type | PSNR↑ | SSIM↑ | LPIPS↓ | FID↓ | PSNR↑ | SSIM↑ | LPIPS↓ | FID↓ | PSNR↑ | SSIM↑ | LPIPS↓ | FID↓ | PSNR↑ | SSIM↑ | LPIPS↓ | FID↓ |
| DPS | *Pixel* | 25.26 | 0.717 | 0.143 | 72.86 | 22.62 | 0.571 | **0.325** | 173.38 | 23.66 | 0.669 | 0.176 | 76.68 | 21.05 | 0.511 | 0.379 | 201.33 |
| ΠGDM | *Pixel* | 26.52 | 0.771 | 0.148 | 81.07 | 23.78 | 0.632 | 0.373 | 169.26 | 26.59 | 0.766 | 0.156 | 83.11 | 23.35 | 0.614 | 0.381 | 191.84 |
| DiffPIR | *Pixel* | 28.07 | 0.771 | 0.177 | 82.78 | 24.59 | 0.619 | 0.515 | 169.36 | 26.95 | 0.696 | 0.206 | 110.89 | 23.61 | 0.548 | 0.433 | 190.47 |
| RED-diff | *Pixel* | 28.37 | 0.823 | 0.237 | 84.94 | 24.91 | 0.679 | 0.498 | 152.17 | 26.63 | 0.773 | 0.219 | 96.76 | 23.56 | 0.633 | 0.458 | 218.74 |
| MGPS | *Pixel* | 27.78 | 0.799 | 0.139 | 67.51 | 24.96 | 0.673 | 0.377 | 140.03 | 26.82 | 0.775 | 0.142 | 72.62 | 23.96 | 0.643 | 0.356 | 156.42 |
| DAPS | *Pixel* | 28.91 | 0.806 | 0.173 | 70.64 | 25.36 | 0.679 | 0.389 | 121.27 | 28.27 | 0.800 | 0.179 | 82.26 | 25.44 | 0.688 | 0.350 | 159.47 |
| AdaPS | *Pixel* | 27.02 | 0.785 | **0.133** | 75.44 | 24.21 | 0.651 | 0.330 | 152.77 | 27.06 | 0.779 | 0.139 | 78.37 | 23.81 | 0.638 | 0.332 | 167.40 |
| ReSample | *Latent* | 25.22 | 0.626 | 0.293 | 105.23 | 22.09 | 0.455 | 0.431 | 183.59 | 24.07 | 0.525 | 0.441 | 119.38 | 20.43 | 0.351 | 0.571 | 219.47 |
| LatentDAPS | *Latent* | 28.50 | 0.826 | 0.265 | 94.70 | 25.13 | 0.685 | 0.469 | 161.20 | 27.58 | 0.791 | 0.245 | 112.06 | 24.37 | 0.646 | 0.455 | 216.88 |
| **SubDPS** | *Dynamic* | 25.45 | 0.718 | 0.155 | 75.30 | 23.12 | 0.586 | 0.407 | 194.30 | 23.78 | 0.672 | 0.186 | 77.53 | 21.38 | 0.528 | 0.461 | 224.00 |
| **SubDAPS** | *Dynamic* | 28.91 | 0.806 | 0.174 | 82.45 | 24.41 | 0.650 | 0.422 | 129.35 | 28.28 | 0.800 | 0.178 | 82.45 | 25.47 | 0.689 | 0.347 | 154.21 |
| **SubDAPS++** | *Dynamic* | 29.17 | 0.828 | 0.162 | 60.38 | 25.59 | 0.704 | 0.393 | 121.60 | 29.36 | 0.822 | 0.115 | 60.43 | 26.25 | 0.721 | 0.274 | 111.67 |
| | | **Nonlinear Deblurring** | | | | | | | | **High Dynamic Range** | | | | | | | |
| Methods | Type | PSNR↑ | SSIM↑ | LPIPS↓ | FID↓ | PSNR↑ | SSIM↑ | LPIPS↓ | FID↓ | PSNR↑ | SSIM↑ | LPIPS↓ | FID↓ | PSNR↑ | SSIM↑ | LPIPS↓ | FID↓ |
| DPS | *Pixel* | 23.10 | 0.653 | 0.212 | 89.66 | 21.24 | 0.518 | 0.438 | 220.63 | 25.65 | 0.709 | 0.128 | 68.19 | 14.40 | 0.431 | 0.542 | 200.85 |
| DiffPIR | *Pixel* | 27.56 | 0.770 | 0.151 | 80.42 | 23.32 | 0.619 | 0.411 | 175.55 | 23.87 | 0.822 | 0.110 | 51.73 | 21.90 | 0.764 | 0.156 | 56.11 |
| RED-diff | *Pixel* | 27.50 | 0.718 | 0.230 | 82.65 | 20.35 | 0.419 | 0.577 | 269.59 | 20.82 | 0.755 | 0.167 | 65.44 | 21.02 | 0.733 | 0.178 | 63.36 |
| MGPS | *Pixel* | 28.05 | 0.806 | **0.115** | 65.73 | 25.31 | 0.700 | 0.265 | 138.59 | **26.46** | 0.807 | **0.092** | 51.58 | 23.63 | 0.747 | 0.175 | 84.15 |
| DAPS | *Pixel* | 27.83 | 0.757 | 0.151 | 79.89 | 25.85 | 0.699 | 0.260 | 125.67 | 25.67 | 0.838 | 0.096 | 50.02 | 23.04 | 0.773 | 0.148 | **54.82** |
| Resample | *Latent* | 24.08 | 0.628 | 0.404 | 126.62 | 22.17 | 0.516 | 0.545 | 235.05 | 25.42 | 0.808 | 0.131 | 71.91 | 24.23 | 0.727 | 0.188 | 99.65 |
| LatentDAPS | *Latent* | 29.29 | 0.838 | 0.169 | 92.15 | 26.49 | 0.755 | 0.256 | 143.84 | 25.49 | 0.810 | 0.163 | 91.65 | 23.03 | 0.706 | 0.274 | 139.62 |
| **SubDPS** | *Dynamic* | 23.14 | 0.651 | 0.214 | 92.33 | 21.50 | 0.527 | 0.444 | 237.26 | 24.78 | 0.784 | 0.143 | 69.48 | 16.17 | 0.464 | 0.502 | 189.16 |
| **SubDAPS** | *Dynamic* | 27.82 | 0.757 | 0.151 | 78.95 | 25.82 | 0.699 | 0.259 | 124.80 | 25.42 | 0.833 | 0.102 | 53.70 | 22.82 | 0.760 | 0.170 | 60.88 |
| **SubDAPS++** | *Dynamic* | 29.76 | 0.838 | 0.115 | 59.58 | 28.08 | 0.793 | 0.173 | 85.58 | 25.52 | 0.839 | 0.095 | 49.05 | 24.30 | 0.784 | 0.146 | 58.25 |

The results presented in Table 1 indicate that SubDPS exhibits a slight performance degradation compared to DPS, whereas SubDAPS achieves nearly the same performance to its full-dimensional versions, DAPS in most cases. This suggests that the adaptation from pixel-space DMs to dynamic resolution DMs incurs only minimal loss in reconstruction fidelity while yielding high computational efficiency improvements, as illustrated in Section 4.2. Additionally, the proposed SubDAPS++ method attains superior performance in the majority of scenarios and remains comparable in the rest. This indicates that the introduction of the threshold parameter $\tau$ to regulate noise injection, combined with the designed corrector, effectively enhances data fidelity. Experimental results for the task of motion deblurring using an image from the ImageNet dataset are illustrated in Figure 2, where the region enclosed by the yellow box indicates that only our proposed method reconstructs the hand of the man on the left. Additional visualization results for other tasks are presented in Appendix J.

## 4.2. Time and Memory Comparison

We evaluate the average inference time per image and the memory usage of these methods using 100 images from the validation set of ImageNet for the nonlinear task of high dynamic range recovery. For methods leveraging dynamic resolution DMs, we report three values for each metric, corresponding to inference at resolutions of $64 \times 64 \times 3$, $128 \times 128 \times 3$, and $256 \times 256 \times 3$. Table 2 indicates that SubDPS, SubDAPS reduce both time and memory usage compared to their full-dimensional counterparts. For example, SubDPS demonstrates an approximately 24% reduction in inference time and reduces memory cost by nearly 77.8% when operating at the $64 \times 64 \times 3$ resolution. Additionally, the proposed SubDAPS++ method achieves lower computational latency than most competing methods. Although SubDAPS++ exhibits inference speeds comparable to RED-diff, it consistently yields higher reconstruction quality across all tasks and metrics.

*Table 2.* Average inference time over 100 images (in seconds) and total memory usage (in MB) for general image restoration methods. With the exception of DPS and SubDPS, all methods utilize 100 NFEs. For methods based on dynamic resolution DMs, we report three values per metric, corresponding to resolutions of $64 \times 64 \times 3$, $128 \times 128 \times 3$, and $256 \times 256 \times 3$ from left to right.

| Methods | DPS | ΠGDM | DiffPIR | RED-diff | MGPS | DAPS | AdaPS | ReSample | LatentDAPS | **SubDPS** | **SubDAPS** | **SubDAPS++** |
|---|---|---|---|---|---|---|---|---|---|---|---|---|
| Time (s) | 122.3 | 12.9 | 9.7 | 7.5 | 110.2 | 16.1 | 18.2 | 74.9 | 53.7 | 92.8 | 14.2 | **7.4** |
| Total Memory Cost (MB) | 8586 | 8620 | 3770 | 13598 | 22166 | 3824 | 9320 | 4756 | 4720 | 3008/4162/8646 | 2784/2856/3890 | 2796/2872/3906 |

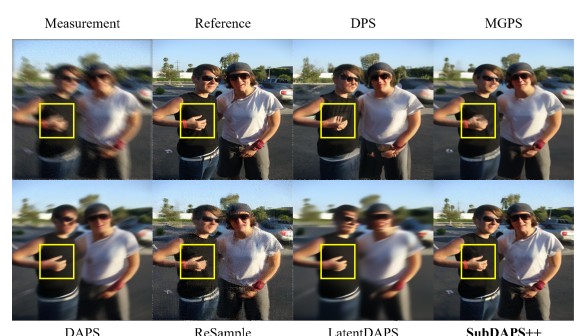

*Figure 2.* Visualization results of our proposed SubDAPS++ method and other baseline methods for the motion deblurring task, with Gaussian noise ($\sigma = 0.05$).

## 5. Conclusion

In this work, we first fine-tune existing pre-trained DMs in pixel space to provide dynamic resolution priors and adapt two full-dimensional DM-based methods, namely DPS and DAPS, to dynamic resolution DMs, resulting in the SubDPS and SubDAPS methods. Additionally, motivated by the favorable performance of SubDAPS, we derive an enhanced variant, denoted as SubDAPS++. Numerical experiments demonstrate both the efficacy and efficiency of the proposed methods.

## Impact Statement

This work employs dynamic resolution DMs to general image restoration tasks, specifically proposing the SubDPS, SubDAPS, and SubDAPS++ frameworks. By projecting data into lower-dimensional subspaces during the sampling process, our methods reduce both computational time cost and memory consumption while maintaining high reconstruction fidelity. Importantly, this work raises no ethical concerns. It aims to improve the efficiency and efficacy of high-quality data restoration without introducing ethical concerns or societal harm.

## Acknowledgment

This work was supported by New Generation Artificial Intelligence-National Science and Technology Major Project (No.2025ZD0123002).

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

# Image Restoration via Dynamic Resolution Diffusion Prior

## Supplementary Material

## A. Detailed Architectural D'iagram of SubDAPS++

We provide a detailed architectural diagram of the proposed SubDAPS++ framework to clarify how the dynamic resolution prior is integrated into the diffusion sampling process.

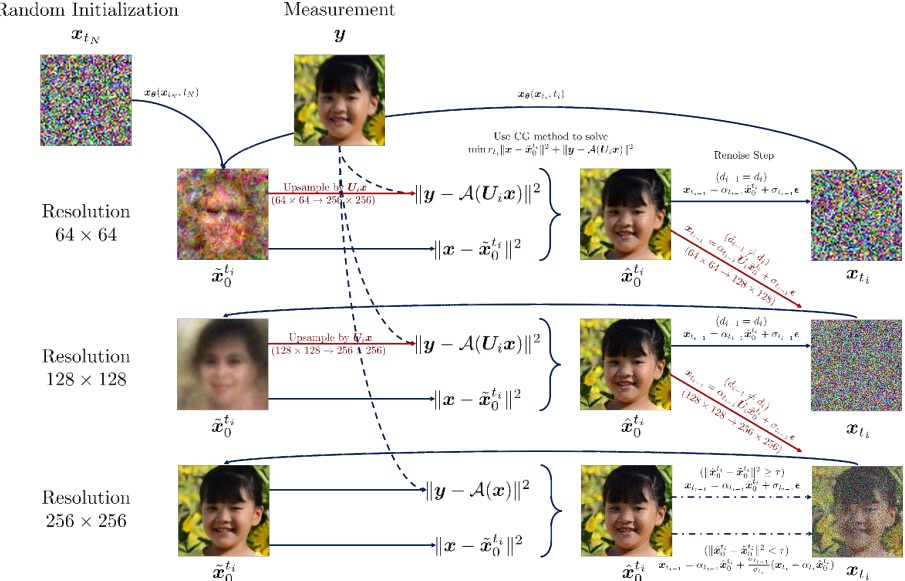

*Figure 3.* Detailed architectural diagram of the proposed SubDAPS++ framework, illustrating how the dynamic resolution prior is integrated into the diffusion sampling loop. Starting from a randomly initialized noisy vector $x_{t_N}$, the restoration process begins in a lower-dimensional subspace and progressively increases the dimensionality throughout sampling. At each resolution, the model first produces an unconditional data estimate $\tilde{x}_0^{t_i}$ using the dynamic resolution diffusion prior. This estimate is then aligned with the measurement through a conjugate gradient procedure, yielding the conditional estimate $\hat{x}_0^{t_i}$. If the dimensionality remains unchanged, the algorithm performs a renoising step to obtain the next noisy sample at the same resolution. Otherwise, the conditional estimate is first upsampled to the next resolution level and then renoised to continue the sampling process. The red arrows indicate the stages involving resolution upsampling. Overall, the figure clarifies the dynamic resolution restoration workflow, in which global structure is recovered at low resolution and finer details are progressively refined at higher resolutions.

## B. SubDAPS++ with existing models

We evaluate SubDAPS++ with off-the-shelf dynamic-resolution DMs and ensembles of resolution-specific pretrained DMs on FFHQ for super-resolution ($4\times$). For the former, we directly use the publicly available dynamic-resolution model provided by Subspace DMs (Jing et al., 2022), and for the latter, we use the publicly available $64 \times 64$, $128 \times 128$, and $256 \times 256$ DMs from the official OpenAI guided-diffusion repository as priors for different resolutions in SubDAPS++. The results in Table 3 below show that both using off-the-shelf dynamic-resolution DMs and using ensembles of resolution-specific pretrained DMs to provide priors at different resolutions yield comparable performance.

*Table 3.* Comparisons of SubDAPS++ on FFHQ **super-resolution** ($4\times$) under different model settings.

| Model Settings | PSNR↑ | SSIM↑ | LPIPS↓ | FID↓ |
|---|---|---|---|---|
| Ensemble of resolution-specific models | 29.41 | 0.839 | 0.162 | 67.68 |
| Off-the-shelf dynamic-resolution model | 28.16 | 0.829 | 0.170 | 119.78 |
| **Fine-tuned model** | 29.34 | 0.838 | 0.157 | 64.61 |

## C. Training Details for Dynamic Resolution Diffusion Models

We utilize pre-trained pixel-space DMs provided by (Chung et al., 2023). The specific training configurations are presented in Table 4.

*Table 4.* Training hyperparameters for the dynamic resolution framework. We report the batch size, learning rate, and total number of iterations employed during fine-tuning.

| Dataset | Batch Size | Learning Rate | Total Iterations |
|---------|-----------|---------------|------------------|
| FFHQ | 4 | $10^{-4}$ | $10^{6}$ |
| ImageNet | 16 | $10^{-5}$ | $2 \times 10^{5}$ |

## D. Illustration of SubDPS, SubDAPS and Conjugate Gradient Method for SubDAPS++

For better understanding, we summarize the complete procedure of SubDPS, SubDAPS and the conjugate gradient method for SubDAPS++ in Algorithms 2, 3 and 4 respectively.

---

**Algorithm 2** SubDPS

---

**Require:** Data prediction model $\boldsymbol{x_\theta}(\cdot, \cdot)$, observation $\boldsymbol{y}$, forward operator $\mathcal{A}(\cdot)$, total number of sampling steps $N$, noise schedule $\{\sigma_{t_i}\}_{i=0}^{N}, \{\alpha_{t_i}\}_{i=0}^{N}$, learned noise parameter $\{\tilde{\sigma}_{t_i}\}_{i=1}^{N}$, guidance strength $\{\zeta_{t_i}\}_{i=1}^{N}$, upsampling matrices $\{\boldsymbol{U}_i\}_{i=1}^{N}$, noise level $\sigma$

1: Sample $\boldsymbol{x}_{t_N} \sim \mathcal{N}(\boldsymbol{0}, \sigma_{t_N}^2 \boldsymbol{I}_{d_N})$
2: **for** $i = N, \ldots, 1$ **do**
3:     $\tilde{\boldsymbol{x}}_0^{t_i} = \boldsymbol{x_\theta}(\boldsymbol{x}_{t_i}, t_i)$
4:     **if** $d_{i-1} = d_i$ **then**
5:        Sample $\boldsymbol{\epsilon} \sim \mathcal{N}(\boldsymbol{0}, \boldsymbol{I}_{d_i})$
6:        $\boldsymbol{x}_{t_{i-1}} = \frac{\alpha_{t_i}}{\alpha_{t_{i-1}}} \frac{\sigma_{t_{i-1}}^2}{\sigma_{t_i}^2} \boldsymbol{x}_{t_i} + \frac{\alpha_{t_{i-1}}}{\sigma_{t_i}^2} \tilde{\boldsymbol{x}}_0^{t_i} + \tilde{\sigma}_{t_i} \boldsymbol{\epsilon}$
7:        $\boldsymbol{x}_{t_{i-1}} = \boldsymbol{x}_{t_{i-1}} - \zeta_{t_i} \nabla_{\boldsymbol{x}_{t_i}} \|\boldsymbol{y} - \mathcal{A}(\boldsymbol{U}_i \tilde{\boldsymbol{x}}_0^{t_i})\|^2$
8:     **else**
9:        Sample $\boldsymbol{\epsilon} \sim \mathcal{N}(\boldsymbol{0}, \boldsymbol{I}_{d_{i-1}})$
10:       $\boldsymbol{x}_{t_{i-1}} = \alpha_{t_{i-1}} \dot{\boldsymbol{U}}_i \tilde{\boldsymbol{x}}_0^{t_i} + \sigma_{t_{i-1}} \boldsymbol{\epsilon}$
11:     **end if**
12: **end for**
13: **Return** $\boldsymbol{x}_{t_0}$

---

## E. Experimental Details

For linear tasks, we first assess the performance of the proposed method using two linear forward operators, namely inpainting (with random $70\%$ masking) and super-resolution ($4\times$ bicubic downscaling). We also examine two linear deblurring tasks, namely Gaussian deblurring and motion deblurring. We employ kernels of size $61 \times 61$ with standard deviations of $3.0$ and $0.5$ for the Gaussian and motion deblurring tasks, respectively. To validate the effectiveness of the proposed method for tasks using nonlinear forward operators, we employ two tasks, namely nonlinear deblurring and high dynamic range recovery. For nonlinear deblurring, learned blurring operators from (Tran et al., 2021) are utilized with a known Gaussian-shaped kernel. All experimental configurations for linear and nonlinear tasks align with those established in prior studies (Chung et al., 2023; Wang et al., 2024; Zhang et al., 2025a). The hyperparameter configurations of SubDAPS++ for different tasks are detailed in Table 5. Here, $N$ represents the total number of sampling steps, $J$ indicates the number of inner iterations, and $\tau$ denotes the threshold parameter for the noise injection strategy.

*Table 5.* Detailed setup of the hyperparameter set in our proposed SubDAPS++ method.

| Hyperparameter | Super-resolution ($4\times$) | Inpainting (random $70\%$) | Gaussian Deblurring | Motion Deblurring | Nonlinear Deblurring | High Dynamic Range |
|---|---|---|---|---|---|---|
| $N$ | 100 | 100 | 100 | 100 | 100 | 100 |
| $J$ | 20 | 20 | 20 | 20 | 100 | 20 |
| $\tau$ | $10^{-5}$ | $10^{-5}$ | $10^{-4}$ | $3 \times 10^{-4}$ | $1 \times 10^{-4}$ | $1 \times 10^{-5}$ |

## F. Comparison between SubDAPS++ and SILO

To further validate the effectiveness of our proposed SubDAPS++, we compare it with SILO, an efficient latent DM-based method that encodes the forward operator to accelerate inference, on Gaussian deblurring, motion deblurring, and super-

---

**Algorithm 3** SubDAPS

---

**Require:** Data prediction model $\boldsymbol{x_\theta}(\cdot,\cdot)$, observation $\boldsymbol{y}$, forward operator $\mathcal{A}(\cdot)$, total number of sampling steps $N$ and inner iterations $J$, noise schedule $\{\sigma_{t_i}\}_{i=0}^{N}, \{\alpha_{t_i}\}_{i=0}^{N}$, regularization parameter $\{r_{t_i}\}_{i=1}^{N}$, step size $\{\eta_{t_i}\}_{i=1}^{N}$, upsampling matrices $\{\boldsymbol{U}_i\}_{i=1}^{N}$, noise level $\sigma$

1: Sample $\boldsymbol{x}_{t_N} \sim \mathcal{N}(\boldsymbol{0}, \sigma_{t_N}^2 \boldsymbol{I}_{d_N})$
2: **for** $i = N, \dots, 1$ **do**
3:     Compute $\tilde{\boldsymbol{x}}_0^{t_i}$ by numerically solving the ODE in Eq. (6) starting from $\boldsymbol{x}_{t_i}$ at timestep $t_i$
4:     $\bar{\boldsymbol{x}}_0^{(0)} = \tilde{\boldsymbol{x}}_0^{t_i}$
5:     **for** $j = 0, \dots J-1$ **do**
6:         Update $\bar{\boldsymbol{x}}_0^{(j+1)} = \bar{\boldsymbol{x}}_0^{(j)} - \eta_{t_i} \nabla_{\bar{\boldsymbol{x}}_0^{(j)}} \mathcal{L}(\bar{\boldsymbol{x}}_0^{(j)}; \tilde{\boldsymbol{x}}_0^{t_i}, t_i) + \sqrt{2\eta_{t_i}}\boldsymbol{z}, \quad \boldsymbol{z} \sim \mathcal{N}(\boldsymbol{0}, \boldsymbol{I}_{d_i})$
7:     **end for**
8:     $\hat{\boldsymbol{x}}_0^{t_i} = \bar{\boldsymbol{x}}_0^{(J)}$
9:     Sample $\boldsymbol{\epsilon}_i \sim \mathcal{N}(\boldsymbol{0}, \boldsymbol{I}_{d_{i-1}})$
10:    $\boldsymbol{x}_{t_{i-1}} = \alpha_{t_{i-1}} \dot{\boldsymbol{U}}_i \hat{\boldsymbol{x}}_0^{t_i} + \sigma_{t_{i-1}} \boldsymbol{\epsilon}_i$
11: **end for**
12: **Return** $\boldsymbol{x}_{t_0}$

---

**Algorithm 4** Conjugate Gradient Method for SubDAPS++

---

**Require:** Data estimate $\tilde{\boldsymbol{x}}_0^{t_i}$ at timestep $t_i$, observation $\boldsymbol{y}$, forward operator $\mathcal{A}(\cdot)$, total number of iterations $J$, regularization parameters $\{r_{t_i}\}_{i=1}^{N}$, upsampling matrices $\{\boldsymbol{U}_i\}_{i=1}^{N}$, noise level $\sigma$

1: $\bar{\boldsymbol{x}}_0^{(0)} = \tilde{\boldsymbol{x}}_0^{t_i}$
2: $\boldsymbol{d}_0 = \boldsymbol{g}_0 = -\nabla_{\bar{\boldsymbol{x}}_0^{(0)}} \mathcal{L}(\bar{\boldsymbol{x}}_0^{(0)}; \tilde{\boldsymbol{x}}_0^{t_i}, t_i)$
3: **for** $j = 0, \dots, J-1$ **do**
4:     $\boldsymbol{\omega}_j = \nabla_{\bar{\boldsymbol{x}}_0^{(j)}} \mathcal{A}(\boldsymbol{U}_i \bar{\boldsymbol{x}}_0^{(j)}) \cdot \boldsymbol{d}_j$
5:     $\alpha_j = (\boldsymbol{g}_j^{\mathrm{T}} \boldsymbol{d}_j) / (r_{t_i} \boldsymbol{d}_j^{\mathrm{T}} \boldsymbol{d}_j + \boldsymbol{\omega}_j^{\mathrm{T}} \boldsymbol{\omega}_j)$
6:     $\bar{\boldsymbol{x}}_0^{(j+1)} = \bar{\boldsymbol{x}}_0^{(j)} + \alpha_j \boldsymbol{d}_j$
7:     $\boldsymbol{g}_{j+1} = -\nabla_{\bar{\boldsymbol{x}}_0^{(j+1)}} \mathcal{L}(\bar{\boldsymbol{x}}_0^{(j+1)}; \tilde{\boldsymbol{x}}_0^{t_i}, t_i)$
8:     $\boldsymbol{d}_{j+1} = \boldsymbol{g}_{j+1} + \frac{\boldsymbol{g}_{j+1}^{\mathrm{T}} \boldsymbol{g}_{j+1}}{\boldsymbol{g}_j^{\mathrm{T}} \boldsymbol{g}_j} \boldsymbol{d}_j$
9: **end for**
10: $\hat{\boldsymbol{x}}_0^{t_i} = \bar{\boldsymbol{x}}_0^{(J)}$
11: **Return** $\hat{\boldsymbol{x}}_0^{t_i}$

---

resolution tasks using the FFHQ dataset. We also report the average runtime computed over 100 images using a single NVIDIA GeForce RTX 4090 GPU. As shown in Table 6, SubDAPS++ consistently achieves higher restoration performance while requiring less inference time than SILO, demonstrating the effectiveness and efficiency of SubDAPS++.

*Table 6.* Comparison between SILO and SubDAPS++ for **Gaussian deblurring**, **motion deblurring** and **super-resolution** tasks on the FFHQ dataset with additive Gaussian noise ($\sigma = 0.05$).

| Methods | Type | Gaussian Deblurring | | | | | Motion Deblurring | | | | | Super-resolution ($4\times$) | | | | |
|---|---|---|---|---|---|---|---|---|---|---|---|---|---|---|---|---|
| | | PSNR↑ | SSIM↑ | LPIPS↓ | FID↓ | Time (s) | PSNR↑ | SSIM↑ | LPIPS↓ | FID↓ | Time (s) | PSNR↑ | SSIM↑ | LPIPS↓ | FID↓ | Time (s) |
| SILO | *Latent* | 26.77 | 0.785 | 0.225 | 78.74 | 122.69 | 20.28 | 0.655 | 0.558 | 178.92 | 255.26 | 26.56 | 0.787 | 0.217 | 74.53 | 255.26 |
| SubDAPS++ | *Pixel* | **29.17** | **0.828** | **0.162** | **60.38** | **7.38** | **29.36** | **0.822** | **0.115** | **60.43** | **7.52** | **29.34** | **0.838** | **0.157** | **64.61** | **5.05** |

# G. Experiments at High Resolution

To validate the effectiveness of SubDAPS++ on high-resolution images, we follow P2L (Chung et al., 2024) and FlowDPS (Kim et al., 2025a) and conduct Gaussian deblurring experiments on DIV2K (Agustsson & Timofte, 2017) at a resolution of $768 \times 768$. The average runtime is computed over 100 images using a single NVIDIA GeForce RTX 4090 GPU. As shown in Table 7, SubDAPS++ achieves superior reconstruction quality while requiring less inference time than P2L and FlowDPS, demonstrating its effectiveness and efficiency in high-resolution image restoration.

*Table 7.* Comparison between P2L, FlowDPS, and SubDAPS++ on **Gaussian deblurring** using the DIV2K dataset at a resolution of $768 \times 768$ with additive Gaussian noise ($\sigma = 0.05$).

| Methods | Type | PSNR↑ | SSIM↑ | LPIPS↓ | FID↓ | Time (s) |
|---|---|---|---|---|---|---|
| P2L | *Latent* | 20.66 | 0.476 | 0.604 | 198.93 | 152.73 |
| FlowDPS | *Latent* | 22.96 | 0.601 | 0.534 | 121.75 | 38.69 |
| SubDAPS++ | *Pixel* | **23.85** | **0.675** | **0.464** | **45.81** | **11.06** |

# H. Ablation Study

In this section, we investigate the impact of the threshold parameter $\tau$ and assess the efficacy of the corrector in the proposed SubDAPS++ method.

## H.1. Ablation Study for Threshold Parameter

We evaluate the effect of the threshold parameter $\tau$ in the proposed SubDAPS++ method on two tasks, namely Gaussian deblurring and motion deblurring. We utilize 100 validation images from the FFHQ dataset for these experiments. We examine five values for $\tau$ from the set $\{3 \times 10^{-4}, 1 \times 10^{-4}, 5 \times 10^{-5}, 1 \times 10^{-5}, 0\}$. The results in Table 8 indicate that the choice of the threshold parameter $\tau$ helps to balance the trade-off between the distortion fidelity and perceptual quality.

*Table 8.* Ablation study on the impact of threshold $\tau$ in the SubDAPS++ method for Gaussian deblurring and motion deblurring tasks using 100 validation images from the FFHQ dataset.

| | Gaussian Deblurring | | | | Motion Deblurring | | | |
|---|---|---|---|---|---|---|---|---|
| Threshold $\tau$ | PSNR↑ | SSIM↑ | LPIPS↓ | FID↓ | PSNR↑ | SSIM↑ | LPIPS↓ | FID↓ |
| $3 \times 10^{-4}$ | 28.95 | 0.816 | 0.147 | 59.11 | 29.36 | 0.822 | 0.115 | 60.43 |
| $1 \times 10^{-4}$ | 29.17 | 0.828 | 0.162 | 60.38 | 29.47 | 0.831 | 0.119 | 59.53 |
| $5 \times 10^{-5}$ | 29.20 | 0.830 | 0.163 | 61.68 | 29.53 | 0.834 | 0.124 | 60.98 |
| $1 \times 10^{-5}$ | 29.25 | 0.833 | 0.172 | 61.42 | 29.57 | 0.838 | 0.135 | 66.51 |
| 0 | 29.42 | 0.843 | 0.203 | 65.14 | 29.76 | 0.844 | 0.151 | 72.43 |

## H.2. Effectiveness of the Corrector

To assess the efficacy of the corrector in Eq. (19), we evaluate the proposed SubDAPS++ method with and without the corrector on the high dynamic range recovery task using 100 validation images from FFHQ and ImageNet datasets. Table 9 demonstrates that incorporating the corrector enhances restoration fidelity.

*Table 9.* Ablation study on the impact of the corrector for the high dynamic range recovery task using 100 validation images from the FFHQ and ImageNet datasets. The notations "w/" and "w/o" denote the presence and absence of the corrector, respectively.

| | FFHQ | | | | ImageNet | | | |
|---|---|---|---|---|---|---|---|---|
| Methods | PSNR↑ | SSIM↑ | LPIPS↓ | FID↓ | PSNR↑ | SSIM↑ | LPIPS↓ | FID↓ |
| SubDAPS++ (w/) | **25.52** | **0.839** | **0.095** | **49.05** | **24.30** | **0.784** | **0.146** | **58.25** |
| SubDAPS++ (w/o) | 25.05 | 0.838 | 0.100 | 50.18 | 23.57 | 0.773 | 0.157 | 59.11 |

# I. Comparison between SubDAPS++ and SubDAPS-F++

A variant of SubDAPS++ that operates at full dimensionality is examined, referred to as SubDAPS-F++. As SubDAPS-F++ does not depend on dynamic resolution priors, it is categorized as a full-dimensional pixel-space method. The performance of SubDAPS-F++ is evaluated for the tasks described in Section 4, including four linear tasks, namely inpainting (random 70%), super-resolution, Gaussian deblurring, and motion deblurring, and two nonlinear tasks, comprising nonlinear deblurring and high dynamic range recovery. These evaluations utilize 100 images sampled from the FFHQ and ImageNet datasets. Results presented in Table 10 indicate that SubDAPS++ achieves performance comparable to SubDAPS-F++ across most metrics, with marginal differences in the remaining cases, indicating the consistency of the proposed approach. Furthermore, the average inference time and memory consumption of SubDAPS-F++ are evaluated on the ImageNet dataset for the high dynamic range recovery task. For SubDAPS-F++, the reconstruction of a single image requires 9.2 seconds and 3966 MB of memory. In contrast, SubDAPS++ requires 7.4 seconds, representing a 19.5% reduction in runtime, and utilizes 2796 MB at the resolution of $64 \times 64 \times 3$, which constitutes a 29.5% reduction in memory usage. The whole procedure of SubDAPS-F++ is summarized in Algorithm 5.

---

**Algorithm 5** SubDAPS-F++

---

**Require:** Data prediction model $\boldsymbol{x_\theta}(\cdot, \cdot)$, observation $\boldsymbol{y}$, forward operator $\mathcal{A}(\cdot)$, total number of sampling steps $N$ and inner iterations $J$,
noise schedule $\{\sigma_{t_i}\}_{i=0}^{N}$, $\{\alpha_{t_i}\}_{i=0}^{N}$, regularization parameter $\{r_{t_i}\}_{i=1}^{N}$, noise level $\sigma$
1: Sample $\boldsymbol{x}_{t_N} \sim \mathcal{N}(\boldsymbol{0}, \sigma_{t_N}^2 \boldsymbol{I}_d)$
2: **for** $i = N, \ldots, 1$ **do**
3:     $\bar{\boldsymbol{x}}_0^{(0)} = \tilde{\boldsymbol{x}}_0^{t_i} = \boldsymbol{x_\theta}(\boldsymbol{x}_{t_i}, t_i)$
4:     $\boldsymbol{d}_0 = \boldsymbol{g}_0 = -\nabla_{\bar{\boldsymbol{x}}_0^{(0)}} \left( r_{t_i} \|\bar{\boldsymbol{x}}_0^{(0)} - \tilde{\boldsymbol{x}}_0^{t_i}\|^2 + \|\boldsymbol{y} - \mathcal{A}(\bar{\boldsymbol{x}}_0^{(0)})\|^2 \right)$
5:     **for** $j = 0, \ldots, J-1$ **do**
6:         $\boldsymbol{\omega}_j = \nabla_{\bar{\boldsymbol{x}}_0^{(j)}} \mathcal{A}(\bar{\boldsymbol{x}}_0^{(j)}) \cdot \boldsymbol{d}_j$
7:         $\alpha_j = (\boldsymbol{g}_j^{\mathrm{T}} \boldsymbol{d}_j) / (r_{t_i} \boldsymbol{d}_j^{\mathrm{T}} \boldsymbol{d}_j + \boldsymbol{\omega}_j^{\mathrm{T}} \boldsymbol{\omega}_j)$
8:         $\bar{\boldsymbol{x}}_0^{(j+1)} = \bar{\boldsymbol{x}}_0^{(j)} + \alpha_j \boldsymbol{d}_j$
9:         $\boldsymbol{g}_{j+1} = -\nabla_{\bar{\boldsymbol{x}}_0^{(j+1)}} \left( r_{t_i} \|\bar{\boldsymbol{x}}_0^{(j+1)} - \tilde{\boldsymbol{x}}_0^{t_i}\|^2 + \|\boldsymbol{y} - \mathcal{A}(\bar{\boldsymbol{x}}_0^{(j+1)})\|^2 \right)$
10:         $\boldsymbol{d}_{j+1} = \boldsymbol{g}_{j+1} + \frac{\boldsymbol{g}_{j+1}^{\mathrm{T}} \boldsymbol{g}_{j+1}}{\boldsymbol{g}_j^{\mathrm{T}} \boldsymbol{g}_j} \boldsymbol{d}_j$
11:     **end for**
12:     $\hat{\boldsymbol{x}}_0^{t_i} = \bar{\boldsymbol{x}}_0^{(J)}$
13:     **if** $\|\hat{\boldsymbol{x}}_0^{t_i} - \tilde{\boldsymbol{x}}_0^{t_i}\|^2 \geq \tau$ **then**
14:         Sample $\boldsymbol{\epsilon}_i \sim \mathcal{N}(\boldsymbol{0}, \boldsymbol{I}_d)$
15:         $\boldsymbol{x}_{t_{i-1}} = \alpha_{t_{i-1}} \hat{\boldsymbol{x}}_0^{t_i} + \sigma_{t_{i-1}} \boldsymbol{\epsilon}_i$
16:     **else**
17:         $\boldsymbol{x}_{t_{i-1}} = \alpha_{t_{i-1}} \hat{\boldsymbol{x}}_0^{t_i} + \frac{\sigma_{t_{i-1}}}{\sigma_{t_i}} (\boldsymbol{x}_{t_i} - \alpha_{t_i} \hat{\boldsymbol{x}}_0^{t_i})$
18:     **end if**
19: **end for**
20: Set $\hat{\boldsymbol{x}}_0^{t_0} = \boldsymbol{x}_{t_0}$ and $\boldsymbol{x}_{t_N}^c = \boldsymbol{x}_{t_N}$
21: **for** $i = N, \ldots, 1$ **do**
22:     $\mathcal{I}_i = \int_{\lambda_{t_{i-1}}}^{\lambda_{t_i}} e^\lambda (\lambda - \lambda_{t_i}) \mathrm{d}\lambda$
23:     $\boldsymbol{x}_{t_{i-1}}^c = \frac{\sigma_{t_{i-1}}}{\sigma_{t_i}} \boldsymbol{x}_{t_i}^c - \left( \sigma_{t_{i-1}} \frac{\alpha_{t_i}}{\sigma_{t_i}} - \alpha_{t_{i-1}} \right) \hat{\boldsymbol{x}}_0^{t_{i-1}} - \sigma_{t_{i-1}} \mathcal{I}_i \frac{\hat{\boldsymbol{x}}_0^{t_{i-1}} - \hat{\boldsymbol{x}}_0^{t_i}}{\lambda_{t_{i-1}} - \lambda_{t_i}}$
24: **end for**
25: **Return** $\boldsymbol{x}_{t_0}^c$

---

*Table 10.* Quantitative evaluation of four linear and two nonlinear tasks under Gaussian noise ($\sigma = 0.05$), using 100 validation images from the FFHQ and ImageNet datasets.

| | | **FFHQ** ($256 \times 256$) | | | | **ImageNet** ($256 \times 256$) | | | | **FFHQ** ($256 \times 256$) | | | | **ImageNet** ($256 \times 256$) | | | |
|---|---|---|---|---|---|---|---|---|---|---|---|---|---|---|---|---|---|
| | | **Inpainting (Random** $70\%$**)** | | | | | | | | **Super-Resolution** ($4\times$) | | | | | | | |
| Methods | Type | PSNR↑ | SSIM↑ | LPIPS↓ | FID↓ | PSNR↑ | SSIM↑ | LPIPS↓ | FID↓ | PSNR↑ | SSIM↑ | LPIPS↓ | FID↓ | PSNR↑ | SSIM↑ | LPIPS↓ | FID↓ |
| **SubDAPS++** | *Dynamic* | 32.21 | 0.907 | 0.056 | 43.15 | 28.61 | 0.843 | 0.092 | 49.15 | 29.34 | 0.838 | 0.157 | 64.61 | 25.79 | 0.722 | 0.358 | 113.91 |
| **SubDAPS-F++** | *Pixel* | 32.17 | 0.907 | 0.056 | 43.24 | 28.55 | 0.843 | 0.092 | 48.99 | 29.34 | 0.838 | 0.157 | 64.62 | 25.83 | 0.722 | 0.356 | 113.84 |
| | | **Gaussian Deblurring** | | | | | | | | **Motion Deblurring** | | | | | | | |
| Methods | Type | PSNR↑ | SSIM↑ | LPIPS↓ | FID↓ | PSNR↑ | SSIM↑ | LPIPS↓ | FID↓ | PSNR↑ | SSIM↑ | LPIPS↓ | FID↓ | PSNR↑ | SSIM↑ | LPIPS↓ | FID↓ |
| **SubDAPS++** | *Dynamic* | 29.17 | 0.828 | 0.162 | 60.38 | 25.59 | 0.704 | 0.393 | 121.60 | 29.36 | 0.822 | 0.115 | 60.43 | 26.25 | 0.721 | 0.274 | 111.67 |
| **SubDAPS-F++** | *Pixel* | 29.17 | 0.828 | 0.162 | 60.40 | 25.62 | 0.705 | 0.392 | 121.45 | 29.36 | 0.822 | 0.115 | 60.39 | 26.28 | 0.722 | 0.273 | 111.15 |
| | | **Nonlinear Deblurring** | | | | | | | | **High Dynamic Range** | | | | | | | |
| Methods | Type | PSNR↑ | SSIM↑ | LPIPS↓ | FID↓ | PSNR↑ | SSIM↑ | LPIPS↓ | FID↓ | PSNR↑ | SSIM↑ | LPIPS↓ | FID↓ | PSNR↑ | SSIM↑ | LPIPS↓ | FID↓ |
| **SubDAPS++** | *Dynamic* | 29.76 | 0.838 | 0.115 | 59.58 | 28.0 | 0.793 | 0.173 | 85.58 | 25.52 | 0.839 | 0.095 | 49.05 | 24.30 | 0.784 | 0.146 | 58.25 |
| **SubDAPS-F++** | *Pixel* | 29.80 | 0.839 | 0.115 | 59.11 | 28.11 | 0.794 | 0.173 | 85.38 | 25.77 | 0.844 | 0.091 | 48.71 | 24.29 | 0.784 | 0.149 | 58.74 |

## J. More Visualization of Experimental Results

We visualize our experimental results on five tasks, namely inpainting (random $70\%$), super-resolution ($4\times$), Gaussian deblurring, motion deblurring. nonlinear deblurring and high dynamic range recovery tasks across two datasets, namely FFHQ and ImageNet. The experiments are conducted under Gaussian noise with a noise level of $\sigma = 0.05$.

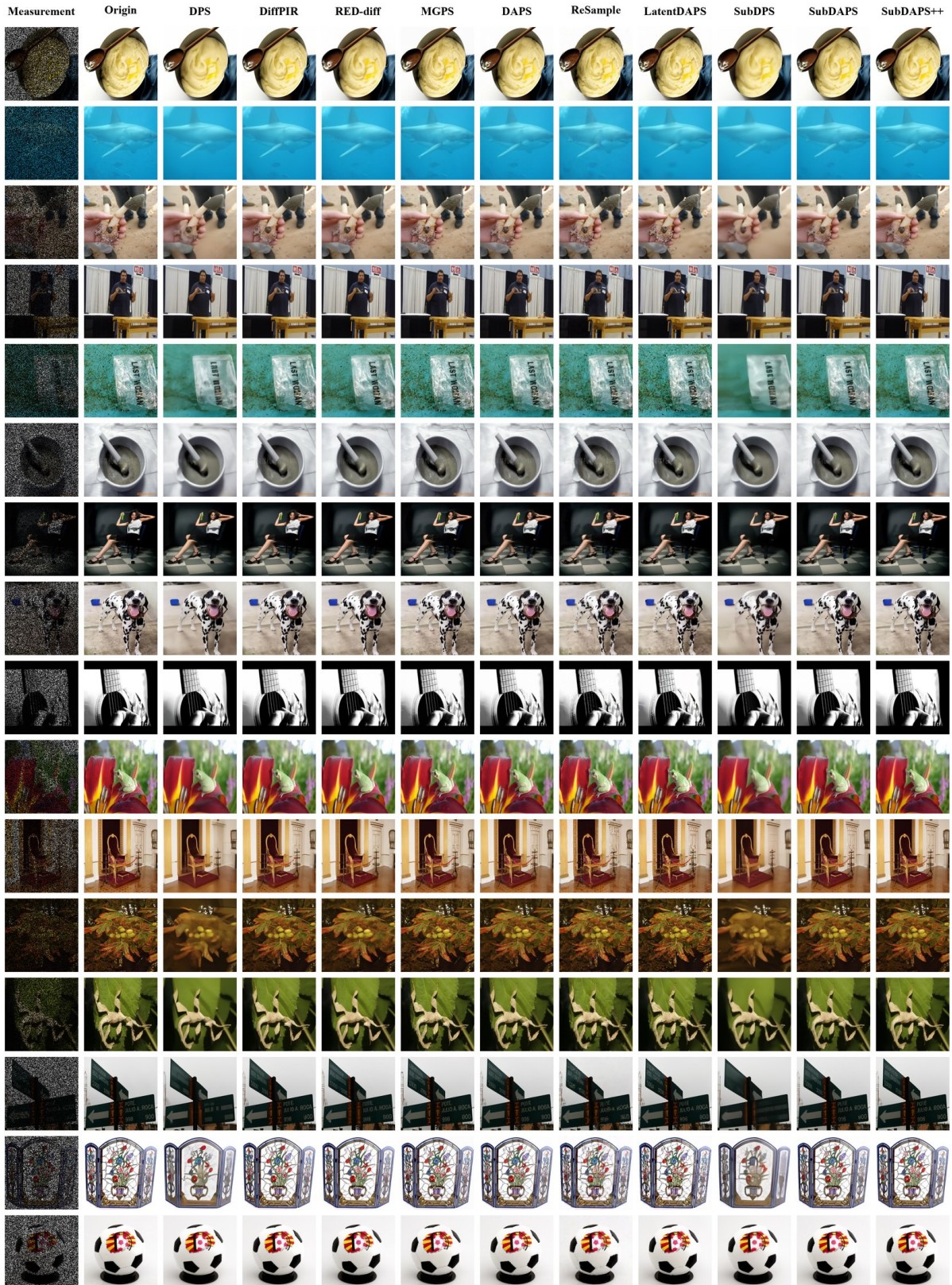

*Figure 4.* Visualization of the experimental results for the inpainting (random 70%) task under Gaussian noise with a noise level of $\sigma = 0.05$.

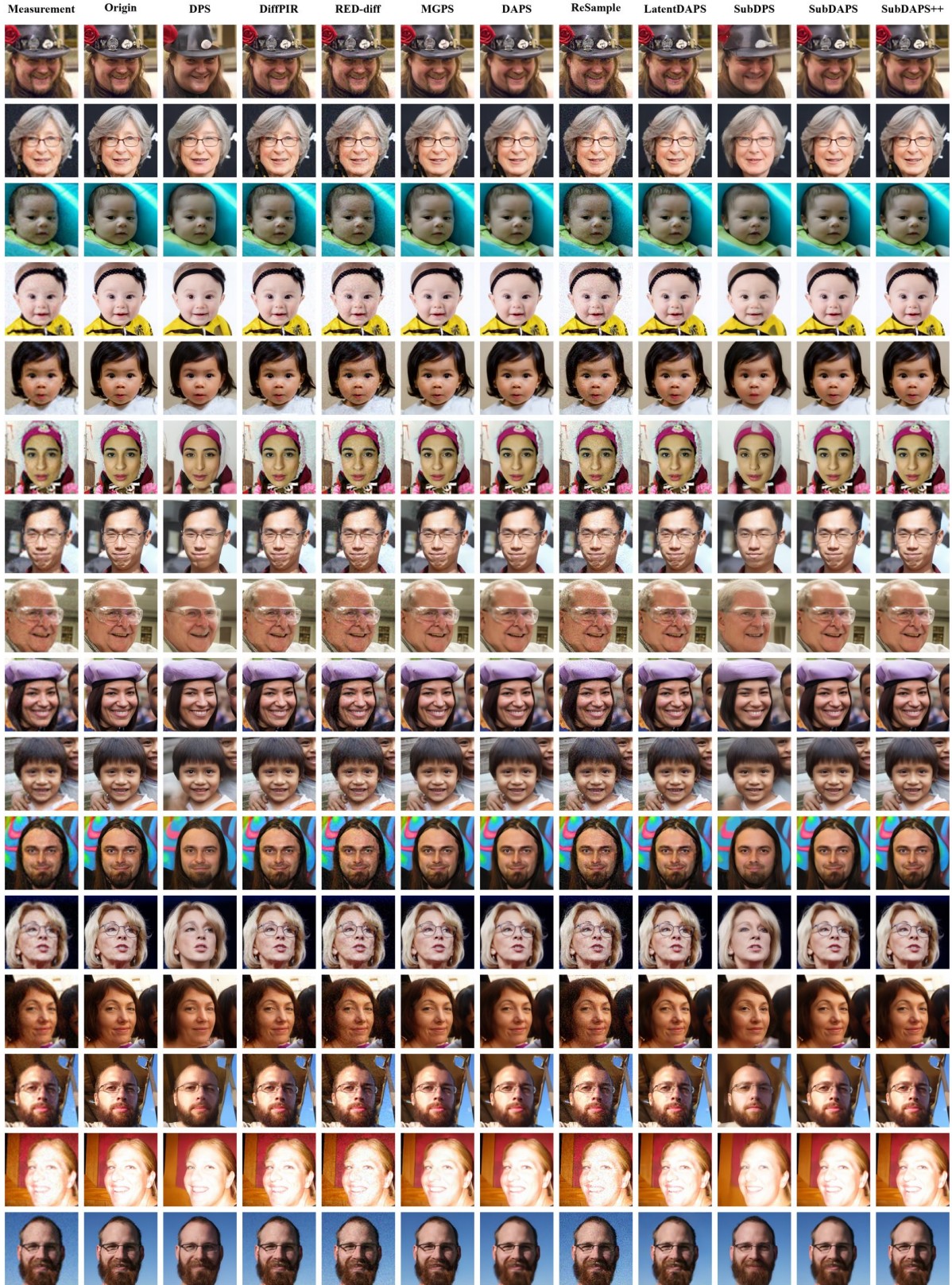

*Figure 5.* Visualization of the experimental results for the super-resolution (4×) task under Gaussian noise with a noise level of $\sigma = 0.05$.

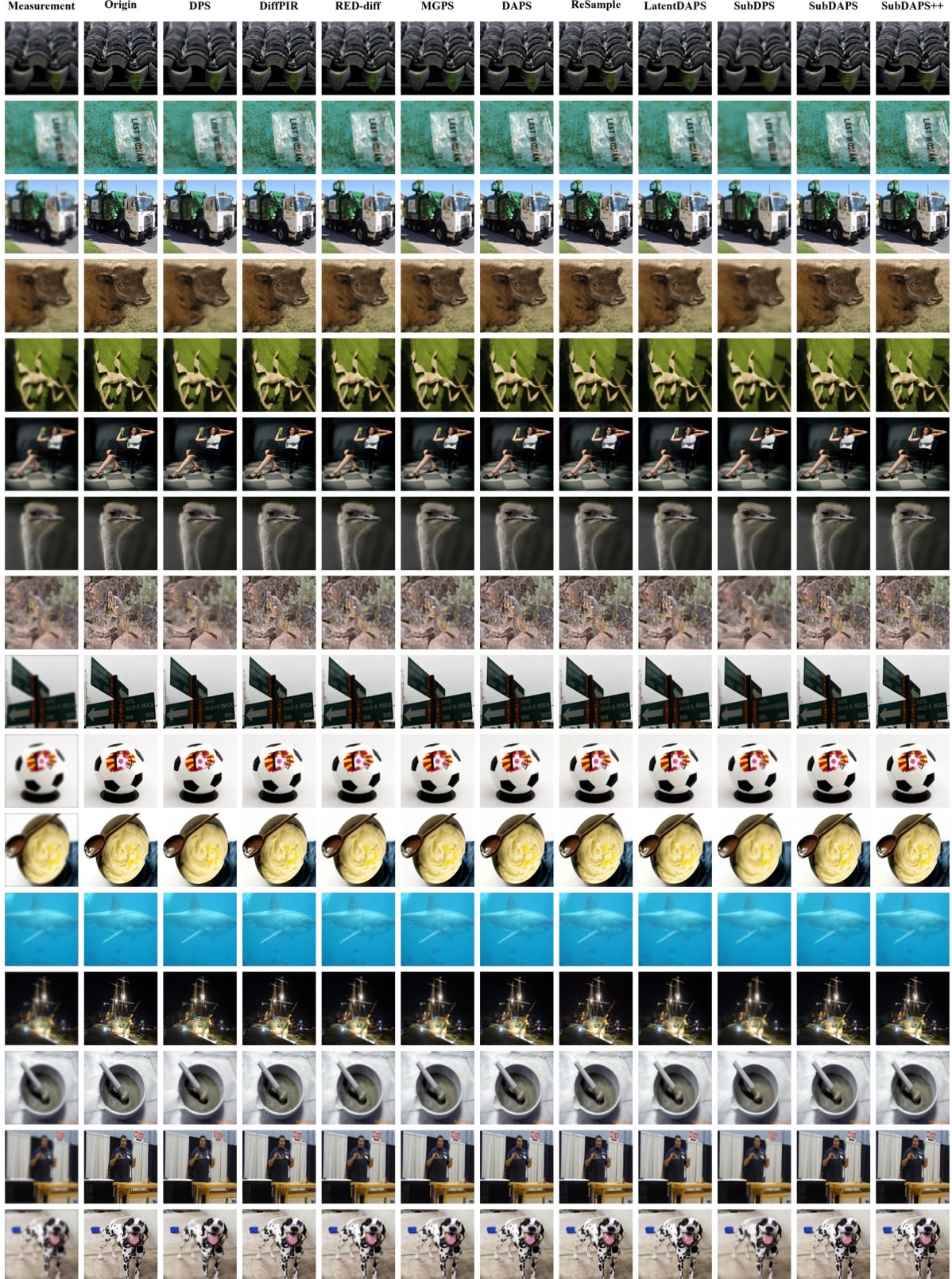

*Figure 6.* Visualization of the experimental results for the Gaussian deblurring task under Gaussian noise with a noise level of $\sigma = 0.05$.

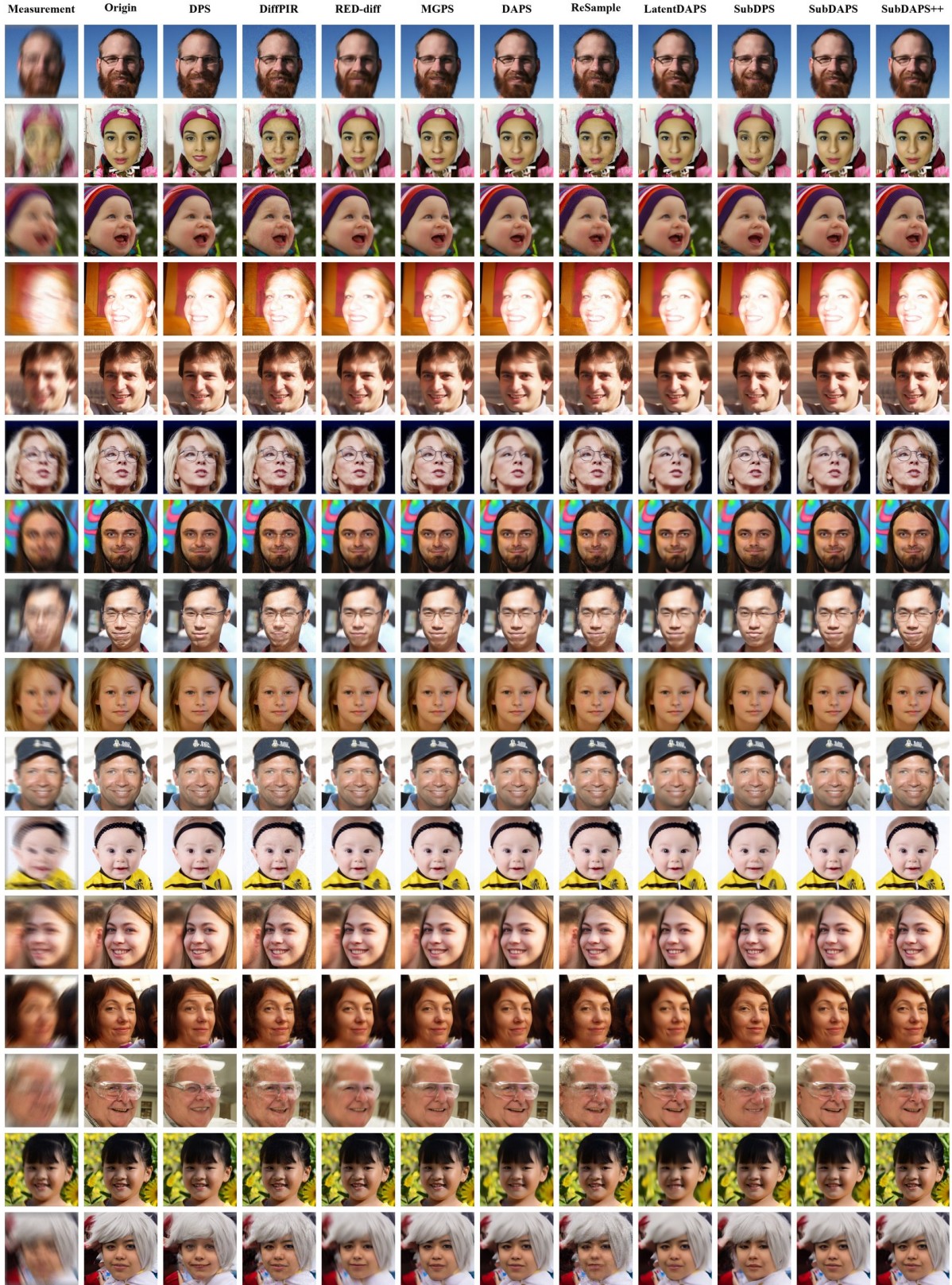

*Figure 7.* Visualization of the experimental results for the motion deblurring task under Gaussian noise with a noise level of $\sigma = 0.05$.

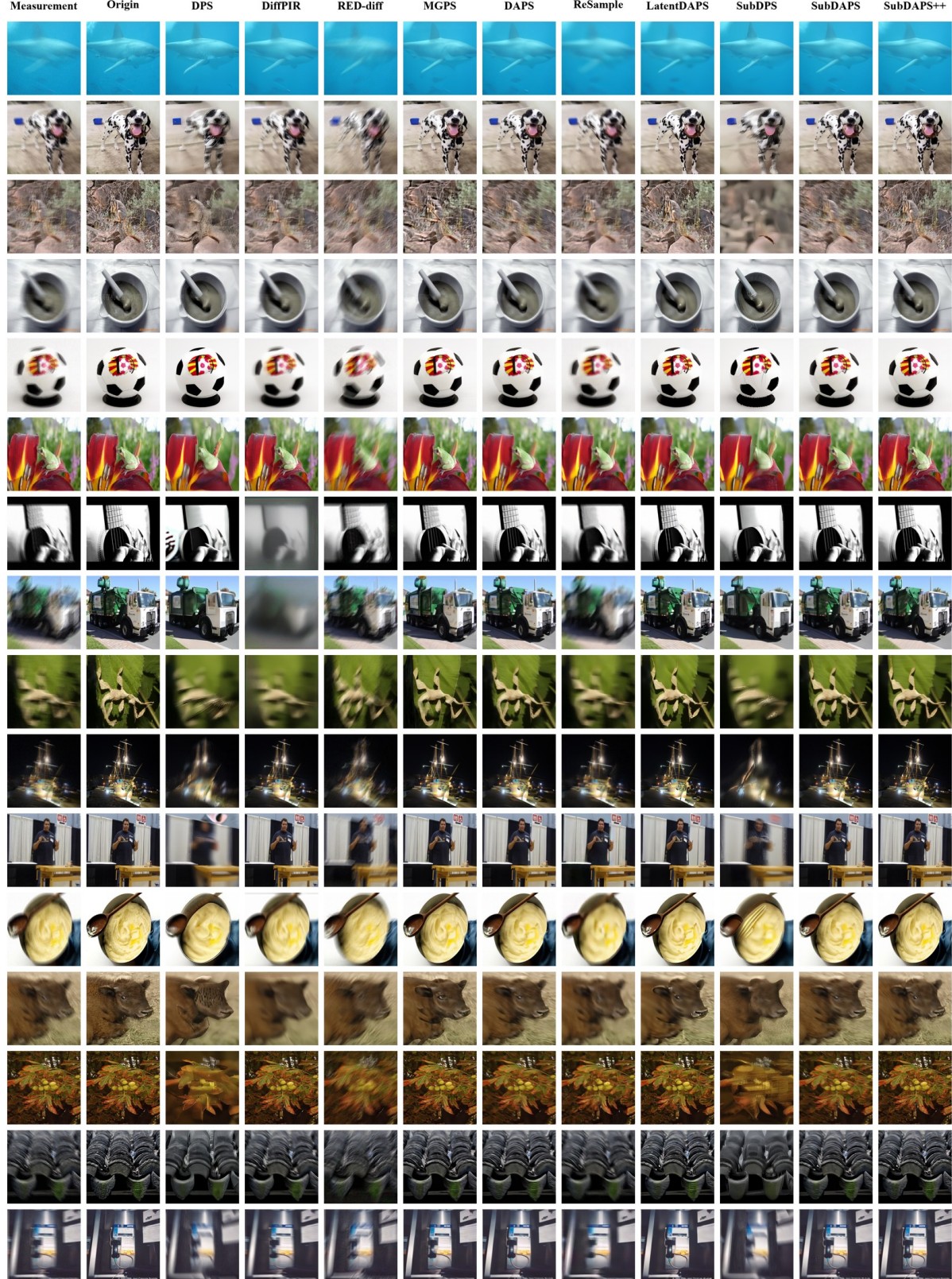

*Figure 8.* Visualization of the experimental results for the nonlinear deblurring task under Gaussian noise with a noise level of $\sigma = 0.05$.

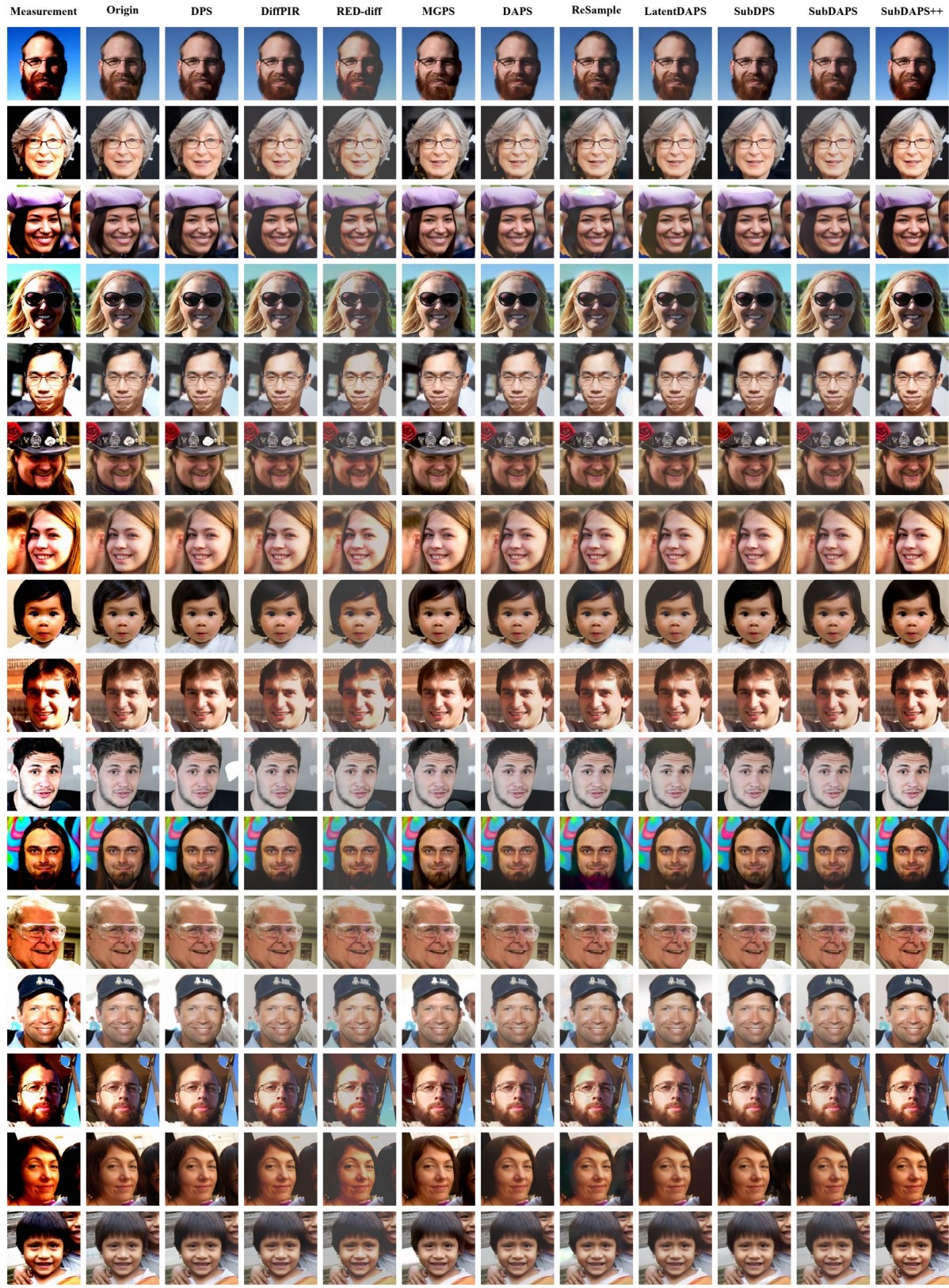

*Figure 9.* Visualization of the experimental results for the high dynamic range recovery task under Gaussian noise with a noise level of $\sigma = 0.05$.

