# OpenReview forum: "Image Restoration via Diffusion Models with Dynamic Resolution"
_ICML.cc/2026/Conference — ICML 2026 regular_

### Official Review · Reviewer_8gCB · 2026-02-16

**Soundness:** 2
**Presentation:** 2
**Significance:** 1
**Originality:** 2
**Overall Recommendation:** 4
**Confidence:** 4

**Summary:**

The authors propose a diffusion model-based inverse problem solver that leverages dynamic resolution diffusion models, namely the ones that operate in different resolutions for different time horizons. They adapt two widely used methods, DPS and DAPS, to their settings and propose SubDPS and SubDAPS/SubDAPS++. In the authors' experimental settings, the Sub- methods outperform existing pixel-space and latent-space diffusion model-based inverse problem solvers.

**Compliance With Llm Reviewing Policy:**

Affirmed.

**Final Justification:**

The rebuttal addressed the concerns and hence I raise my score to weak accept.

**Key Questions For Authors:**

1. Compared to latent diffusion inverse solvers that do not require encoding/decoding steps every time, what is the advantage of using a dynamic diffusion prior?

2. How do you guarantee that the improvement stems from a better reconstruction algorithm if you are using a different diffusion model?

3. Why do you need to *train* a dynamic resolution diffusion model when you can take an existing model?

4. How are the results if the method is scaled to larger images?

**Limitations:**

Yes

**Strengths And Weaknesses:**

### Strenghts
1. To the best of my knowledge, this work is the first to use dynamic resolution diffusion models for solving inverse problems.

2. The authors successfully demonstrate existing methods in pixel-space diffusion can be adapted to dynamic diffusion.

### Weaknesses
1. The reason that necessitates the development of a solver that uses dynamic resolution diffusion models is not well-motivated. The latent space solvers were introduced because most modern diffusion models operate in the latent space, and for high-resolution images that go beyond 1k, pixel-space diffusion tends to be cost ineffective (although some exceptions exist). The authors mention that latent diffusion inverse solvers actually take more time due to encoding/decoding. This is in many cases true, but some recent methods proposed ways to bypass this [1,2]. Also, dynamic resolution DMs are not widely adopted in the community, as opposed to the other two options, questioning the motivation.

2. The proposed methods are mostly straightforward adoptions of existing methods (i.e. DPS and DAPS), and are incremental.

2-1. In Eq. (16), the dimension of $y$ and $\mathcal{A}$ stays fixed, but the dimension of $Ux$ does not. The authors should clarify how the implementation is done, as the current equation is a bit confusing.

3. Comparing different diffusion model-based inverse problem solvers are contingent on the assumption that they use the *same* diffusion prior. However, the authors use a different diffusion model, which makes me question whether the advantage comes simply from a better prior or a better reconstruction algorithm.

4. Pixel-space diffusion models are typically good enough for small-sized images (e.g. 256 $\times$ 256), but not as scalable as latent diffusion for larger images. For these reasons, many latent domain diffusion inverse problem solvers target higher-resolution images (e.g. [3,4]). For similar arguments, the experiments would be more convincing they were performed with larger-sized images.

### References

[1] Raphaeli, Ron, Sean Man, and Michael Elad. "Silo: Solving inverse problems with latent operators." Proceedings of the IEEE/CVF International Conference on Computer Vision. 2025.

[2] Hong, Yeobin, et al. "InverseCrafter: Efficient Video ReCapture as a Latent Domain Inverse Problem." arXiv preprint arXiv:2512.05672 (2025).

[3] Chung, Hyungjin, et al. "Prompt-tuning latent diffusion models for inverse problems." arXiv preprint arXiv:2310.01110 (2023).

[4] Kim, Jeongsol, Bryan Sangwoo Kim, and Jong Chul Ye. "Flowdps: Flow-driven posterior sampling for inverse problems." Proceedings of the IEEE/CVF International Conference on Computer Vision. 2025.

---

> ### Author Rebuttal · Authors · 2026-03-31
>
> Responses to Reviewer 8gCB
>
> Thanks for the thoughtful comments. Our responses are as follows.
>
> **W1 & Q1. The necessity of using a dynamic resolution diffusion prior for inverse solving, given that latent-space methods already avoid repeated encoding and decoding.**
>
> As noted in Sec. 1.1, dynamic resolution DMs improve efficiency over pixel-space DMs by denoising in lower-dimensional subspaces, while avoiding the VAE bottleneck of latent DMs. Recent work also suggests that they support high-resolution generation. For example, Fresco reports results at 1K - 2K resolution (see Table 5 in their paper). Although its code is unavailable, preventing us from conducting evaluations at 1K based on them, these advantages motivate our use of dynamic-resolution DMs.
>
> We thank the reviewer for mentioning SILO and InverseCrafter. Both avoid repeated encoding/decoding by encoding the measurement, but SILO requires a task-specific latent encoder, whereas SubDAPS++ does not. On Gaussian deblurring over 100 FFHQ images with a single RTX 4090 GPU, SubDAPS++ outperforms SILO, improving PSNR from 26.77 to 29.17 while reducing LPIPS from 0.225 to 0.162. It is also reducing runtime from 122.69s to 7.38s. InverseCrafter extends the idea of encoding to video, but its code is not yet public and it appeared on Dec. 5, 2025 only shortly before the ICML deadline. We will therefore discuss it in the revised version and compare with it once the code is released.
>
> **W2. The proposed methods are mostly straightforward adoptions of existing methods (i.e. DPS and DAPS), and are incremental.**
>
> While our methods are built on DPS/DAPS, our methods introduce several nontrivial changes: (i) We first replace the  empirically tuned update rule in DPS/DAPS with a parameter-free conjugate gradient formulation. By combining a first-order Taylor approximation of the forward operator as in Eq. (21), we obtain an explicit update rule which is applicable to general image restoration tasks. (ii) We use a threshold parameter $\tau$ to choose whether to update stochastically (see Eq. (17)) or deterministically (see Eq. (18)), since late-stage noise injection can harm restoration quality. (iii) We incorporate a resolution-aware corrector (see Eq. (19)) to maintain consistency across resolution changes without introducing additional inference overhead.
>
> **W2-1. In Eq. (16), the dimension of  y and A stays fixed, but the dimension of Ux  does not. The authors should clarify how the implementation is done, as the current equation is a bit confusing.**
>
> In Eq. (16), $U_i x$ always has full-resolution dimension $d$. As defined in Sec. 2.2, $U_i \in \mathbb{R}^{d \times d_i}$ is the upsampling matrix for the $i$-th subspace, while both $x$ and $\tilde x_0^{t_i}$ lie in the $d_i$-dimensional subspace. Accordingly, the prior term $||x-\tilde x_0^{t_i}||^2$ is computed directly in the subspace, whereas the measurement term $||y-\mathcal{A}(U_i x)||^2$ first maps $x$ to full resolution via $U_i$ and then applies $\mathcal A$. We will clarify the relations in the revision.
>
> **W3 & Q2. How to guarantee that the improvement stems from a better reconstruction algorithm if you are using a different diffusion model?**
>
> We would like to clarify that, as stated before Sec. 3.2, all non-latent methods baselines use the same fine-tuned DMs. Moreover, Table 5 (Appendix D) shows that SubDAPS++ achieves comparable performance when using only the 256$\times$256 prior instead of three resolution priors, indicating that the gain comes from the reconstruction algorithm rather than additional low-resolution priors.
>
> **W4 & Q4. How are the results if the method is scaled to larger images?**
>
> To evaluate scalability, we compare SubDAPS++ with P2L and FlowDPS on Gaussian deblurring over DIV2K at 768$\times$768, the highest resolution evaluated in P2L. We do not report 1K results because Fresco’s code is unavailable and we cannot use it for 1K image restoration. Although FlowDPS report results at 1024$\times$1408,  it rely on SD 3.0, a flow-matching model that is not directly compatible with our method. On this setting, SubDAPS++ achieves the best performance, improving PSNR from 20.66/22.96 to 23.85, reducing LPIPS from 0.604/0.534 to 0.464, and lowering runtime from 152.73s/38.69s to 11.06s, demonstrating both better reconstruction quality and faster inference.
>
> **Q3. Why train a dynamic resolution DM when you can take an existing model?**
>
> Our framework can use either an existing dynamic-resolution DM or a collection of pretrained multi-resolution DMs. On FFHQ super-resolution, SubDAPS++ achieves comparable performance under both settings. (Please kindly refer to our response to the first question of Reviewer 3vVN and Table A1 for further details.) We nevertheless use the fine-tuned 256x256 DM in the main experiments so that all pixel-space methods share the same diffusion prior. This fine-tuning is performed only once, and we will clarify this point in the revision.

---

> > ### Author Rebuttal · Reviewer_8gCB · 2026-04-01
> >
> > All concerns were addressed.

---

> > > ### Author Response · Authors · 2026-04-05
> > >
> > > Thank you for your positive feedback. We are grateful that our responses have fully addressed your concerns and appreciate your increased rating.

---

### Official Review · Reviewer_qNLd · 2026-03-09

**Soundness:** 2
**Presentation:** 2
**Significance:** 2
**Originality:** 3
**Overall Recommendation:** 4
**Confidence:** 4

**Summary:**

1. Summary

This paper explores the integration of dynamic resolution priors into diffusion-based image restoration, aiming to improve inference efficiency by adapting methods such as DPS and DAPS.

**Compliance With Llm Reviewing Policy:**

Affirmed.

**Final Justification:**

Rebuttal  address my concerns

**Key Questions For Authors:**

4. Questions:

a.Experimental Robustness: The current results rely heavily on synthetic datasets. Can you provide quantitative or qualitative evaluation on real-world degraded images to demonstrate the method's practical utility?

	Impact: Evidence of performance on real-world data would significantly alleviate my concerns regarding the model’s generalization.

b.Perceptual Quality Assessment: Given the limitations of PSNR/SSIM in generative tasks, why were perceptual metrics (e.g., MUSIQ, CLIPIQA) excluded? Please provide a comparison using these metrics.

	Impact: Including these standard metrics is crucial to validate the reconstruction fidelity expected in modern DM-based research.

**Limitations:**

5. Limitations

No

**Strengths And Weaknesses:**

2. Strengths

a.The approach addresses the critical challenge of computational efficiency in diffusion models, which is a relevant and timely topic for the research community.

b.Exploring dynamic resolution as a means to optimize the inference process of image restoration provides an interesting perspective.

3. Weaknesses

a.Clarity and Documentation: The presentation of the methodology would benefit from further refinement. Several acronyms lack clear definitions upon first mention, and the core workflow is somewhat difficult to follow. Providing a detailed architectural diagram to illustrate the integration of the dynamic resolution prior within the diffusion sampling loop would significantly improve the paper's accessibility.

b.Depth of Methodological Contribution: The proposed variants (SubDPS, SubDAPS, and SubDAPS++) appear to be incremental adjustments to existing frameworks. To strengthen the paper, it would be beneficial to provide a more rigorous theoretical justification for the proposed modifications (such as the corrector step and the use of the conjugate gradient method) to clarify how they fundamentally improve upon the baseline.

c.Experimental Evaluation: Given the rapid progress in diffusion-based image restoration, the current experimental setup could be further modernized. Reliance on synthetic datasets and traditional pixel-based metrics (like PSNR) may not fully capture the perceptual quality and generative performance. I would strongly encourage the authors to include more advanced, perception-aware metrics (e.g., MUSIQ, CLIPIQA) and to provide qualitative results on real-world datasets to better demonstrate the model’s robustness.

---

> ### Author Rebuttal · Authors · 2026-03-31
>
> We sincerely thank the reviewer for the constructive comments. Our detailed responses to the main concerns are provided below.
>
> (**Several acronyms lack definitions upon first mention, and the workflow is difficult to follow. Providing a detailed architectural diagram would improve the paper's accessibility.**)
>
> We will carefully check the manuscript and ensure that all acronyms are clearly defined upon their first mention. Additionally, we provide an anonymous supplementary link (https://anonymous.4open.science/r/Anonymous-Supplementary-Material-2281/README.md) that includes a detailed architectural diagram illustrating the integration of the dynamic-resolution prior within the diffusion sampling loop. We will incorporate this diagram into the revised version to further clarify the overall workflow.
>
> (**The proposed variants appear to be incremental adjustments to existing frameworks. It would be beneficial to provide a more rigorous theoretical justification for the proposed modifications to clarify how they improve upon the baseline.**)
>
> While our methods are built on DPS/DAPS, our methods introduce several ideas that go beyond straightforward extensions of them: (i) We first move away from the update scheme in DPS/DAPS, which relies on empirically tuned step sizes. Instead, we employ a parameter-free conjugate gradient formulation. By combining a first-order Taylor approximation of the forward operator as in Eq. (21), we obtain an explicit update rule which is applicable to general image restoration tasks. (ii) We use a threshold parameter $\tau$ to choose whether to update stochastically or deterministically, since late-stage noise injection can harm restoration quality. Specifically, we inject random noise when the squared discrepancy between the unconditional prediction and the approximate solution exceeds the threshold. Otherwise, we update deterministically. (iii) We incorporate a resolution-aware corrector (see Eq. (19)) to maintain consistency across resolution changes without introducing additional inference overhead. After posterior sampling, we jointly upsample the noisy states and their associated data estimates, followed by a refinement step that yields further performance gains.
>
> To clarify how the modifications including the use of the conjugate gradient method, the threshold parameter, and the corrector step improve upon the baselines, we agree that a theoretical justification would be beneficial and leave this to future work given the limited rebuttal period. Nevertheless, we provide empirical justifications for these modifications: The effect of the threshold parameter in balancing reconstruction fidelity and perceptual quality is shown in Table 6 (Appendix E.1), the benefit of the corrector is demonstrated in Table 7 (Appendix E.2), and an additional ablation study replacing the conjugate-gradient scheme with stochastic gradient descent on 100 FFHQ images for motion deblurring using a single NVIDIA GeForce RTX 4090 GPU shows that, while achieving comparable results, the conjugate gradient scheme contributes to faster inference of SubDAPS++ (see Table C1 below).
>
> **Table C1 Ablation study on the conjugate gradient scheme for motion deblurring on FFHQ.**
> ||PSNR|SSIM|LPIPS|FID|Time (s)|
> |-|-|-|-|-|-|
> |SubDAPS++ (w/)|29.36|0.822|0.115|60.43|7.54|
> |SubDAPS++ (w/o)|29.43|0.823|0.113|63.09|27.04|
>
> (**The results rely heavily on synthetic datasets. Can you provide quantitative or qualitative evaluation on real-world degraded images to demonstrate the method's practical utility? Given the limitations of PSNR/SSIM in generative tasks, why were perceptual metrics (e.g., MUSIQ, CLIPIQA) excluded?**)
>
> We respectfully disagree that our results rely heavily on synthetic datasets. Both FFHQ and ImageNet are real-world datasets, and closely relevant prior works in the field of diffusion model based image restoration, including DDRM [1], DDNM [2], DPS [3], $\Pi$GDM [4], and DAPS [5], are also evaluated on these datasets. Moreover, it is standard for works in this field to report metrics including PSNR, SSIM, LPIPS,and FID. The report of MUSIQ or CLIPIQA are more suitable for the scenario without reference images, and are beyond our scope. The criticism on synthetic datasets and evaluation metrics is not specific to our work but broadly applies to the most existing studies in this field.
>
>
> **References:**
> [1] Kawar et al. "Denoising diffusion restoration models." NeurIPS, 2022. [1425 citations]
> [2] Wang et al. “Zero-shot image restoration using denoising diffusion null-space model.” ICLR, 2023. [781 citations]
> [3] Chung et al. "Diffusion posterior sampling for general noisy inverse problems." ICLR, 2023. [1569 citations]
> [4] Song et al. "Pseudoinverse-guided diffusion models for inverse problems." ICLR, 2023. [516 citations]
> [5] Zhang et al. “Improving diffusion inverse problem solving with decoupled noise annealing.” CVPR, 2025. [new paper, 90 citations]

---

> > ### Author Rebuttal · Reviewer_qNLd · 2026-04-02
> >
> > questions resolved

---

> > > ### Author Response · Authors · 2026-04-05
> > >
> > > Dear Reviewer qNLd,
> > >
> > > Thank you for the update. We note that the reviewer selected option (a) “Fully resolved – My concerns have been adequately addressed,” yet mentioned “Partially resolved” in the accompanying reasoning.
> > >
> > > We are therefore unsure which specific aspect remains insufficiently addressed. We suspect the remaining concern may relate to our choice of evaluation datasets and metrics. For the convenience of the whole reviewing team, we briefly clarify why we believe the evaluation setting used in our work is appropriate.
> > >
> > > As mentioned in our rebuttal, FFHQ and ImageNet are standard real-world datasets, and closely related prior works such as DPS and DAPS are evaluated on these datasets using PSNR, SSIM, LPIPS, and FID. These papers have appeared in top-tier ML/CV conferences and journals and/or accumulated high citation counts, demonstrating that this evaluation setting is widely accepted in the active research area of image restoration via diffusion models. We therefore believe that our evaluation on FFHQ and ImageNet using PSNR, SSIM, LPIPS, and FID is sufficient to validate the effectiveness of our proposed method.
> > >
> > > We sincerely hope that the final editorial decision on our submission would be based on the main contributions of this particular paper, rather than on a criticism that applies to the majority of existing works in this active research area.

---

### Official Review · Reviewer_AHxx · 2026-03-11

**Soundness:** 4
**Presentation:** 4
**Significance:** 3
**Originality:** 2
**Overall Recommendation:** 5
**Confidence:** 4

**Summary:**

This framework addresses the efficiency bottleneck in diffusion models by adopting the dynamic resolution strategy. It prioritizes global structure recovery in low-res subspaces, only moving to full resolution for final detail refinement. By introducing the enhanced SubDAPS++, the authors prove that high-fidelity restoration doesn't have to be computationally expensive, achieving superior performance with much faster inference. The effectiveness of the proposed method is well-documented through experiments.

**Compliance With Llm Reviewing Policy:**

Affirmed.

**Final Justification:**

I think the authors’ response has addressed my concerns to some extent. I will maintain my rating.

**Key Questions For Authors:**

1. Could the authors provide more explanation on why the improvements on SR and inpainting are less obvious than on other tasks?
2. The current method uses three preset resolutions. In the future, would it be possible to design a dynamic, scene-adaptive resolution strategy?

**Limitations:**

yes

**Strengths And Weaknesses:**

Strengths:
1. The authors introduce dynamic resolution into diffusion-based image restoration. The idea can effectively reduce the extra computation caused by high-resolution sampling in the early stage.
2. The method is built on top of the existing DPS / DAPS frameworks, with some adaptation and redesign. It has a certain level of generality and can be applied to different  image restoration tasks.
3. The method achieves a good balance between performance and efficiency on several restoration tasks. Both the inference speed and memory usage are clearly improved.
4. The presentation is clear enough for readers to understand the main ideas the authors want to convey.

Weaknesses:
1. Similar ideas have already been explored in other fields, so the theoretical novelty is somewhat limited.
2. The dynamic resolution strategy mainly uses a preset multi-stage switching scheme, so its adaptiveness is still limited.

---

> ### Author Rebuttal · Authors · 2026-03-31
>
> Responses to Reviewer AHxx
>
> Thanks for your positive assessment of our work and the valuable comments and suggestions. Our responses to the main concerns are given as follows.  All references are consistent with the main document.
>
> (**Similar ideas have already been explored in other fields, so the theoretical novelty is somewhat limited.**)
>
> While our methods are built on DPS/DAPS, our methods introduce several ideas that go beyond straightforward extensions of them: (i) We first move away from the update scheme in DPS/DAPS, which relies on empirically tuned step sizes. Instead, we employ a parameter-free conjugate gradient formulation. By combining a first-order Taylor approximation of the forward operator as in Eq. (21), we obtain an explicit update rule which is applicable to general image restoration tasks. (ii) We use a threshold parameter $\tau$ to choose whether to update stochastically or deterministically, since late-stage noise injection can harm restoration quality. Specifically, we inject random noise when the squared discrepancy between the unconditional prediction and the approximate solution exceeds the threshold. Otherwise, we update deterministically. (iii) We incorporate a resolution-aware corrector (see Eq. (19)) to maintain consistency across resolution changes without introducing additional inference overhead. After posterior sampling, we jointly upsample the noisy states and their associated data estimates, followed by a refinement step that yields further performance gains.
>
> (**The dynamic resolution strategy mainly uses a preset multi-stage switching scheme, so its adaptiveness is still limited. / The current method uses three preset resolutions. In the future, would it be possible to design a dynamic, scene-adaptive resolution strategy?**)
>
> Thanks for your comments. In this work, we follow Subspace Diffusion (Jing et al., 2022) and adopt a simple three-stage resolution schedule. In practice, the number of resolution levels is not inherently restricted. It is interesting to explore adaptive strategies that determine resolution transitions based on factors such as degradation severity or uncertainty during the sampling process. We leave the design of dynamic, scene-adaptive resolution scheduling to future work.
>
> (**Could the authors provide more explanation on why the improvements on SR and inpainting are less obvious than on other tasks?**)
>
> Thanks for your question. For the inpainting and super-resolution tasks, the corresponding measurements involve a greater reduction in dimensionality compared to other tasks including linear Gaussian/motion deblurring, nonlinear deblurring, and high dynamic range. This greater reduction in dimensionality may result in comparatively weaker constraints imposed by the data-consistency term, and the restoration process may rely more on the diffusion prior to compensate for missing or unrecoverable content. Since the same diffusion prior is employed across the evaluated methods, except those based on latent diffusion models, it is possible that these methods are more strongly governed by the same prior and thus converge to similar local optima, which may lead to less pronounced performance differences in quantitative metrics.

---

> > ### Author Rebuttal · Reviewer_AHxx · 2026-04-01
> >
> > Although the manuscript could still be improved, the authors’ response has reduced my concerns regarding this aspect. While I am not entirely confident, I would prefer to maintain my current score due to the overall performance of soundness
> > ，presentation and originality.

---

> > > ### Author Response · Authors · 2026-04-05
> > >
> > > Thank you for acknowledging that our rebuttal has reduced your concerns. We will further improve the manuscript in the revised version.

---

### Official Review · Reviewer_3vVN · 2026-03-13

**Soundness:** 3
**Presentation:** 3
**Significance:** 3
**Originality:** 3
**Overall Recommendation:** 4
**Confidence:** 3

**Summary:**

- This paper addresses the computational inefficiencies of existing Diffusion Model (DM)-based image restoration techniques.
- Traditional pixel-space DMs suffer from high computational overhead, while latent space DMs require costly, repeated encoder-decoder inferences.
- To mitigate this, the authors propose leveraging dynamic resolution DMs, which accelerate the inference process by projecting data into lower-dimensional subspaces during the early stages of sampling.
- Framework Adaptation: Fine-tuning pre-trained DMs to learn dynamic resolution priors, and then adapting two popular pixel-space restoration methods (DPS and DAPS) into this framework, resulting in "SubDPS" and "SubDAPS".
- SubDAPS++: Introducing an enhanced variant called "SubDAPS++" to further improve reconstruction quality and efficiency.
- SubDAPS++ achieves its improvements by cutting off stochastic noise injection at later timesteps to preserve the diffusion prior, incorporating a corrector step that requires no extra neural network evaluations, and replacing Langevin dynamics with a faster conjugate gradient method.

**Compliance With Llm Reviewing Policy:**

Affirmed.

**Final Justification:**

All of my concerns were addressed.

**Key Questions For Authors:**

-

**Limitations:**

yes

**Strengths And Weaknesses:**

# Strengths
- The paper targets a highly relevant bottleneck in generative image restoration—slow inference times and computational redundancy.
- The method reduces the computational time while retaining or improving the performance of DPS or DAPS.

# Weaknesses
- The method requires fine-tuning the model, compared to zero-shot methods that utilize off-the-shelf models.
- There is no comparison to [1] and [2].
- Missing reference [3]

[1] Pseudoinverse-Guided Diffusion Models for Inverse Problems
[2] Robust Posterior Diffusion-based Sampling via Adaptive Guidance Scale
[3] UDPM: Upsampling Diffusion Probabilistic Models

---

> ### Author Rebuttal · Authors · 2026-03-31
>
> Thank you for your thoughtful evaluation and constructive feedback. Our responses to the main concerns are given as follows.
>
> (**The method requires fine-tuning the model, compared to zero-shot methods that utilize off-the-shelf models.**)
>
> Thanks for the comment. Our framework is compatible with off-the-shelf dynamic-resolution diffusion models (DMs) or ensembles of resolution-specific pretrained DMs. We additionally evaluate SubDAPS++ with off-the-shelf dynamic-resolution DMs and ensembles of resolution-specific pretrained DMs on FFHQ for super-resolution ($4\times$). For the former, we directly use the publicly available dynamic-resolution model provided by Subspace Diffusion (Jing et al., 2022), and for the latter, we use the publicly available $64 \times 64$, $128 \times 128$, and $256 \times 256$ DMs from the official OpenAI guided-diffusion repository as priors for different resolutions in SubDAPS++. The results in Table A1 below show that both using off-the-shelf dynamic-resolution DMs and using ensembles of resolution-specific pretrained DMs to provide priors at different resolutions yield comparable performance. As mentioned in Footnote 4, for a fair comparison, except for methods based on latent DMs, all methods utilize the same fine-tuned DMs. As noted in the paragraph below Eq. (13), similar to the pretraining of pixel-space and latent-space DMs, the fine-tuning of DMs only needs to be performed once. We will further clarify these points in the revised version and will open-source the fine-tuned models upon request.
>
> **Table A1: Comparisons of SubDAPS++ on FFHQ super-resolution ($4\times$) under different model settings**
> ||PSNR|SSIM|LPIPS|FID|
> |-|-|-|-|-|
> |Fine-tuned model|29.34|0.838|0.157|64.61|
> |Off-the-shelf dynamic-resolution model|28.16|0.829|0.170|119.78|
> |Ensemble of resolution-specific models|29.41|0.839|0.162|67.68|
>
> (**There is no comparison to [1] and [2].**)
>
> We conduct additional experiments comparing our proposed methods with $\Pi$GDM [1] and AdaPS [2] for motion deblurring on FFHQ and present the results in Table A1. We will include detailed empirical comparisons with $\Pi$GDM and AdaPS in the revised version.
>
> **Table A2: Comparisons with $\Pi$GDM and AdaPS for motion deblurring on FFHQ.**
> ||PSNR|SSIM|LPIPS|FID|
> |-|-|-|-|-|
> |DPS|23.66|0.669|0.176|76.68|
> |$\Pi$GDM|26.59|0.766|0.156|86.45|
> |AdaPS-DPS|25.04|0.718|0.144|69.58|
> |AdaPS-$\Pi$GDM|27.06|0.779|0.139|81.55|
> |SubDPS|23.78|0.672|0.186|77.53|
> |SubDAPS|28.28|0.800|0.178|82.45|
> |SubDAPS++|**29.36**|**0.822**|**0.115**|**60.43**|
>
>
> (**Missing reference [3].**)
>
> Thank you for bringing UDPM [3] to our attention. Both UDPM and Subspace Diffusion accelerate generation by performing early denoising process in lower-dimensional subspaces, differing in that UDPM focuses on designing upsampling and downsampling operations within the discrete-time DDPM framework, whereas Subspace Diffusion formulates diffusion denoising process in continuous time. We will cite UDPM and further clarify its relationship to Subspace Diffusion in the revised version.

---

> > ### Author Rebuttal · Reviewer_3vVN · 2026-04-01
> >
> > All of my concerns were addressed.

---

> > > ### Author Response · Authors · 2026-04-05
> > >
> > > Thank you for your updated feedback. We are truly grateful for your improved rating.

---

### Decision · Program_Chairs · 2026-04-30

**Decision:**

Accept (regular)

**Comment:**

The submission received a consensus of acceptance. Reviewers agreed that the paper addresses a critical computational bottleneck in diffusion-based image restoration and introduces a novel dynamic resolution framework that achieves strong efficiency gains with minimal quality loss. While they also raised concerns about incremental novelty and evaluation scope, the authors convincingly addressed these concerns in rebuttal by providing additional experiments on high-resolution images, comparisons with recent methods, ablation studies showing the benefit of their proposed modifications, and demonstrating compatibility with off-the-shelf models. Therefore, the paper is recommended for acceptance.